# *SFPQ-TFE3* reciprocally regulates mTORC1 and induces lineage plasticity in a mouse model of renal tumorigenesis

Kaushal Asrani [1,2,10] ✉, Adrianna Amaral [1,10], Juhyung Woo[1], Sanaz Nourmohammadi Abadchi[1], Thiago Vidotto[1], Eddie Imada[2], Alyza Skaist[3], Kewen Feng[3], Hans B. Liu[1], Mithila Kasbe[1], Yorifumi Satou [4], Masaya Baba [5], Yuichi Oike [6], Patricia Outeda[7], Terry Watnick[7], Avi Z. Rosenberg[1], Laura S. Schmidt [8,9], W. Marston Linehan [8], Pedram Argani[1] & Tamara L. Lotan [1,2,3] ✉

MiT/TFE gene fusions like *SFPQ-TFE3* drive both epithelial (translocation RCC) and mesenchymal (PEComas) neoplasms. However, no mouse models for *SFPQ-TFE3*-related tumors exist and the underlying mechanisms of lineage plasticity remain unclear. Here, we demonstrate that constitutive murine renal expression of *SFPQ-TFE3* disrupts kidney development with early neonatal renal failure and death, while post-natal induction induces infiltrative epithelioid tumors, that morphologically and transcriptionally resemble human PEComas, with strong activation of mTORC1 signaling via increased V-ATPase expression. Remarkably, *SFPQ-TFE3* expression is sufficient to induce lineage plasticity, with down-regulation of the PAX2/PAX8 nephric lineage factors and tubular epithelial markers, and up-regulation of PEComa differentiation markers in transgenic mice, cell lines and human tRCC. mTOR inhibition down-regulates *SFPQ-TFE3* expression and rescues PAX8 expression and transcriptional activity in vitro. These data provide evidence of an epithelial cell-of-origin for *TFE3*-driven PEComas, highlighting a reciprocal role for *SFPQ-TFE3* and mTOR in driving lineage plasticity in the kidney.

Translocation renal cell carcinoma (tRCC) is a rare subtype of non-clear cell, sporadic kidney cancer driven by chromosomal translocations involving the MiT/TFE family of transcription factors (*TFE3* [Xp11.23], *TFEB* [6p21.1], and *MITF* [3p13]), which are key regulators of lysosomal biogenesis. The resulting fusions of MiT/TFE genes with various partner genes (the most common being *ASPSCR1, PRCC*, and *SFPQ*), leads to constitutive nuclear localization and activation of the chimeric transcription factors[1,2]. tRCC is a common renal carcinoma subtype occurring in children, representing up to a third of all cases, and though rarer in adults, it is frequently aggressive and more common in females. Notably, tRCC exhibit a high degree of morphologic and clinical heterogeneity, at least in part due to varying *TFE3* fusion

[1]Department of Pathology, Johns Hopkins University School of Medicine, Baltimore, MD, USA. [2]Department of Urology, Johns Hopkins University School of Medicine, Baltimore, MD, USA. [3]Department of Oncology, Johns Hopkins University School of Medicine, Baltimore, MD, USA. [4]Division of Genomics and Transcriptomics, Kumamoto University, Kumamoto, Japan. [5]Department of Urology, Graduate School of Medical Sciences, Kumamoto University, Kumamoto, Japan. [6]Department of Molecular Genetics, Graduate School of Medical Sciences, Kumamoto University, Kumamoto, Japan. [7]University of Maryland, School of Medicine, Baltimore, MD, USA. [8]Urologic Oncology Branch, Center for Cancer Research, National Cancer Institute, National Institutes of Health, Bethesda, MD, USA. [9]Basic Science Program, Frederick National Laboratory for Cancer Research, Frederick, MD, USA. [10]These authors contributed equally: Kaushal Asrani, Adrianna Amaral. ✉e-mail: kasrani1@jhmi.edu; tlotan1@jhmi.edu

partners[2], and their molecular landscape has only been partially defined, prompting an urgent need to identify biomarkers and therapeutic targets.

In addition to renal carcinomas, *TFE3*-fusions are also implicated in the pathogenesis of a subset of potentially aggressive mesenchymal tumors called perivascular epithelioid cell tumors (PEComas), which are derived from an unknown cell-of-origin, and most frequently occurring in the kidney in addition to other sites such as bladder and colon[3]. Like tRCC, *TFE3*-fusion associated PEComas also tend to arise in young adults, are more common in females and may be associated with prior chemotherapy[3]. tRCC and PEComas have highly overlapping epithelioid morphologies and immunophenotypes, including expression of melanocytic lineage genes and lysosomal markers (PMEL, MelanA, Cathepsin K, GPNMB) that are canonical MiT/TFE transcriptional targets[4–6]. Pathologically, the key distinction between tRCC and MiT/TFE-driven renal PEComas is the lack of apparent epithelial differentiation in the latter, as evidenced by the complete absence of pan-epithelial markers (cytokeratins/EMA/CD10)[7] and loss of critical renal lineage transcription factors, PAX8 and PAX2[8–10]. In contrast, there is retention of these renal epithelial lineage markers in tRCC, similar to what is seen in clear cell RCC, where PAX8 is essential for oncogenesis[11,12].

Though some have posited that PEComas originate from pericytes[13] or neural crest cells[14,15], at least one study has indicated a potential clonal origin of renal PEComa cells from tumor-initiating, renal proximal tubular epithelial cells[16] and hybrid epithelial/mesenchymal tumors have been described in patients[17]. This fact, together with the frequent underexpression of cytokeratin and EMA expression seen in tRCC cases[18], suggests the possibility that tRCC and renal PEComas may develop from the same epithelial cell-of-origin, which subsequently undergoes transdifferentiation in the latter. However no in vivo transgenic models of PEComa tumorigenesis have been described to date, hampering progress in understanding mechanisms of tumorigenesis and potential therapies for this rare, but potentially aggressive, tumor type.

Though rearrangements involving the MiT/TFE genes *TFE3* and *TFEB* are critical drivers for a subset of PEComas, a majority of renal PEComas (including the most common and well-differentiated subtype known as angiomyolipomas) are driven by alternate genomic alterations, most commonly biallelic *TSC1/2* loss resulting in mTORC1 hyperactivation. Explaining the mutual exclusivity between *TSC1/2* inactivation and MiT/TFE gene rearrangements in PEComas, we and others have previously shown that mTORC1 activation leads to constitutive activation of *TFEB* and *TFE3* through *FLCN* inactivation[19,20]. Accordingly, inactivating alterations in *FLCN* also drive a rare subset of PEComas, in addition to renal carcinomas that bear resemblance to tRCC[21–23]. Of note, the mTOR pathway is itself strongly activated in tRCC via largely undefined mechanisms; multiple studies have shown enrichment of mTOR signaling in human tRCC[18,24–26], as well as cell-line[27], PDX[28], and one transgenic mouse tRCC model[26]. Recent studies in renal carcinoma systems have underscored the reciprocity of this mTORC1-MiT interaction in the kidney: in the setting of *TSC1/2* or *FLCN* loss, TFEB and/or TFE3 are constitutively activated, drive renal tumorigenesis and these transcription factors are also upstream drivers of mTOR signaling[19,20,29,30], indicating that the persistent co-activation of catabolic (autophagy and lysosomal biogenesis) and anabolic (mTORC1) transcriptional programs is a hallmark - albeit paradoxical - feature of many types of renal tumors.

Here, we describe a transgenic mouse model of renal tumorigenesis induced by *SFPQ-TFE3* expression, the most common *TFE3* gene fusion seen in human PEComas and their melanotic variants[3,10,31], and the last of the three most common tRCC fusions that has yet to be modeled in mice. We demonstrate that *SFPQ-TFE3* induces the development of renal tumors recapitulating human PEComas when constitutively expressed in postnatal renal tubular epithelial cells, constituting a transgenic model system for this rare tumor type. Remarkably, *SFPQ-TFE3* expression is sufficient to rapidly drive lineage plasticity in renal tubular cells, evidenced by downregulation of PAX2/8 expression, loss of cytokeratin expression, and melanocytic/lysosomal marker upregulation. MiT/TFE fusion-driven renal tumorigenesis is accompanied by early activation of the mTORC1 pathway across multiple model systems, which requires MiT/TFE-mediated V-ATPase gene expression. Highlighting the persistently bi-directional regulation of MiT/TFE activity and mTOR, mTOR signaling inhibition is sufficient to rescue renal lineage marker expression and transcriptional activity by downregulating *TFE3*-fusion expression. Taken together, these transgenic models highlight a distinct form of epithelial lineage plasticity driven by *TFE3*-fusions and implicate the critical role of mTOR signaling in MiT/TFE fusion-driven tumor types.

## Results

### Generation of an inducible murine allele of SFPQ-TFE3

We generated a transgenic mouse which conditionally expresses the human *SFPQ-TFE3* fusion, downstream of a *LoxP-Stop-LoxP (LSL)* cassette, under control of a strong chicken beta-actin (CAG) promoter in the Rosa 26 locus (*SFPQ-TFE3^LSL*; hereafter referred to as *ST* mice). Briefly, the *LSL-SFPQ-TFE3* allele, containing the human *SFPQ* CDS (exon 1-9) /human *TFE3* CDS (exon 5-10) (type 2 fusion)[1], was cloned into intron 1 of the *Rosa26* locus in reverse orientation, separated from the native CAG promoter by a stop sequence flanked by *LoxP* sites (*LSL*), thus enabling conditional excision of the stop sequence upon Cre-mediated recombination and expression of the fusion transgene (Supplementary Fig. 1A). To validate conditional expression of the allele, renal tubular epithelial cells from adult C57BL/6 controls or *ST* transgenic mice were harvested, dissociated, and cultured in the presence of a control adenovirus or adenovirus expressing Cre-recombinase. Primary cells from *ST*, but not control mice, showed strong induction of SFPQ-TFE3 expression and the canonical MiT/TFE E-box target GPNMB[4] on immunoblotting (Supplementary Fig. 1B). Genotypes were additionally confirmed by PCR (Supplementary Fig. 1C, D).

### Ksp-Cadherin Cre-mediated induction of SFPQ-TFE3 disrupts kidney development with renal failure and early neonatal death

To conditionally express *SFPQ-TFE3* in tubular epithelial cells during renal development, we crossed *ST* transgenic mice with Ksp-Cadherin (Cadherin-16 [Cdh16]) -Cre mice, in which the Cre recombinase is specifically expressed from the Cdh16 promoter in distal tubular epithelial cells and collecting ducts of the kidney starting from embryonic day E12.5[32], to generate *SFPQ-TFE3^LSL; Ksp-Cre* mice (hereafter referred to as *STK* mice). While the *STK* mice were born in expected mendelian ratios and did not manifest any gross anatomic differences compared to littermate controls at birth, no *STK* mice survived beyond postnatal day 20 (P20) (Fig. 1A). Histologically, *STK* kidneys had a disorganized appearance at P1 with mild and scattered tubular dilation and abundant expression of nuclear TFE3 in tubular epithelial cells by IHC (Fig. 1B). By P15, *STK* kidneys were grossly enlarged and markedly cystic, with disruption of the cortical architecture (Fig. 1C, D), and significantly elevated kidney to body weight ratios, as well as blood urea nitrogen (BUN) and serum creatinine (Fig. E–G). Indirect immunofluorescence for LTL, which labels the brush border of differentiated proximal tubules, showed diminished staining in the *STK* kidney tubules, compared to littermate controls, with rare, dilated tubules expressing distal tubular markers such as DBA (Fig. 1H). Histological analysis of kidneys from 3-week-old *STK* mice demonstrated tubular enlargement by solid nests of large epithelioid cells with abundant clear-to-eosinophilic cytoplasm and monomorphic nuclei (Fig. 1I). The mutant kidneys also showed luminal occlusion with frequent psammomatous calcification (Fig. 1I), a feature of human tRCC[6], and

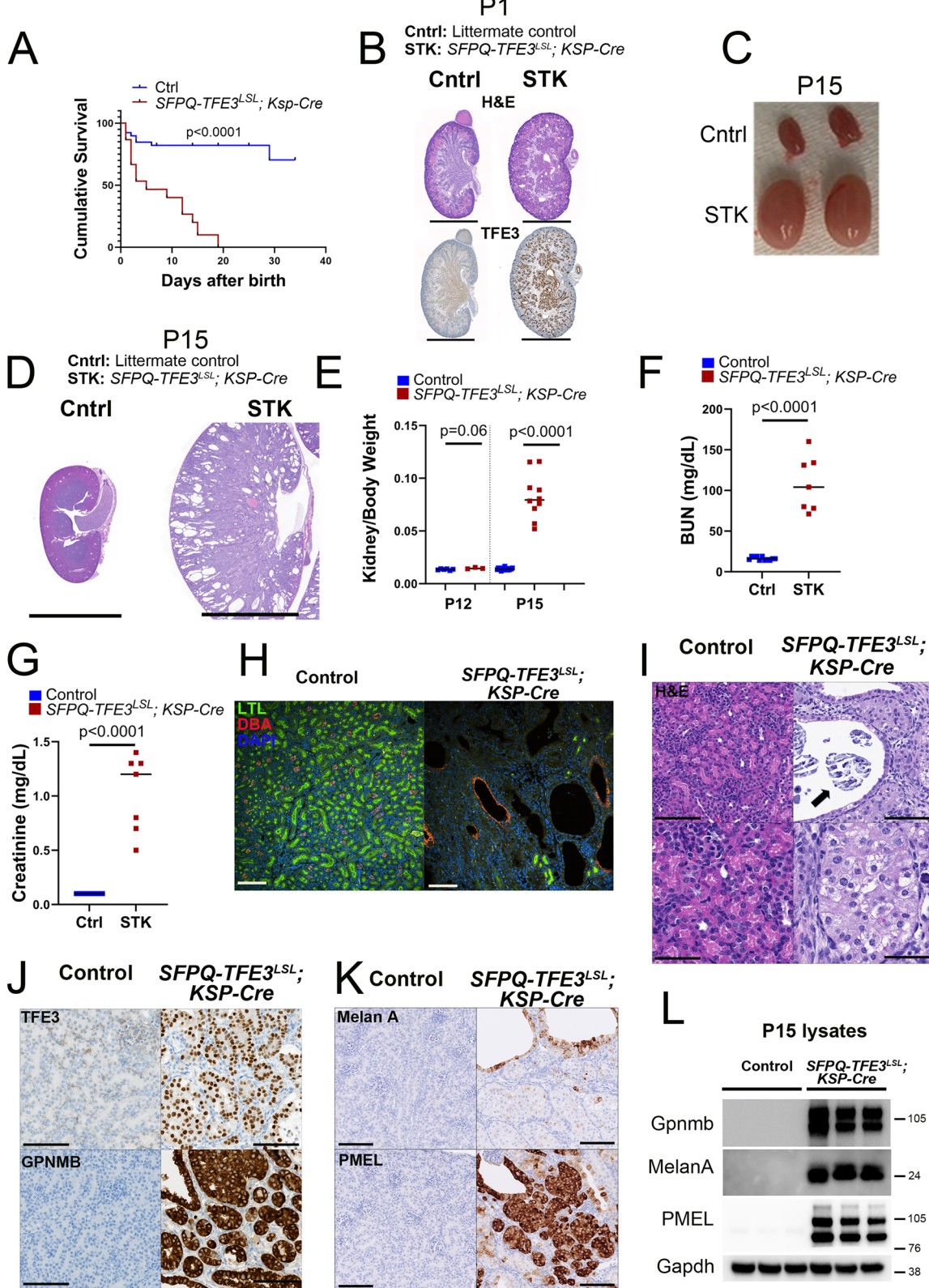

trichrome staining highlighted inter-tubular fibrosis (Supplementary Fig. 2A). By P15, there was diffuse tubular nuclear TFE3 expression in *STK* kidneys, and upregulation of an array of melanocytic/lysosomal markers over-expressed in human MiT/TFE-related neoplasms (PMEL, MelanA and GPNMB)[3,4,6] (Fig. 1J–L). Consistent with a pro-tumorigenic role of SFPQ-TFE3, proliferation as measured by BrDU positivity, Ki67 staining, and phosphorylated H3 were all significantly elevated in

*STK* kidneys compared to littermate controls by 2 weeks of age (Supplementary Fig. 2B–D). Cumulatively, these results indicate that neonatal expression of renal *SFPQ-TFE3* disrupts renal development with subsequent renal failure culminating in early postnatal death. Though there was evidence of increased tubular cell proliferation, the early renal failure precluded aging mice to assess tumor development in the *STK* model.

**Fig. 1 | Ksp-Cadherin Cre-mediated induction of SFPQ-TFE3 disrupts kidney development with renal failure and early neonatal death. A** Kaplan-Meier survival analyses of age-matched, control and *SFPQ-TFE3^LSL; Ksp-Cre* transgenic mice. The following numbers were analyzed: [control = 39, STK = 15]. Data presented as median, p < 0.0001 by Log-rank (Mantel-Cox) test. **B** Hematoxylin and eosin (H&E) and TFE3 IHC staining of kidneys from age-matched (1 day old), control and *SFPQ-TFE3^LSL; Ksp-Cre* transgenic mice. Scale bar: 1 mm. **C** Macroscopic images of kidneys from age-matched (15 days old), control and *SFPQ-TFE3^LSL; Ksp-Cre* transgenic mice. **D** H&E images of kidneys from age-matched (15 days old), control and *SFPQ-TFE3^LSL; Ksp-Cre* transgenic mice. Scale bar: 5 mm. **E** Kidney to body weight ratios of age-matched, control and *SFPQ-TFE3^LSL; Ksp-Cre* transgenic mice on post-natal day 12 and day 15. The following numbers were analyzed: a) Kidneys to body weight ratios at day 12 [control = 6, STK = 3] and day 15 [control = 17 and STK = 10]. Data presented as median, p-values by two-tailed, Mann-Whitney test. **F** BUN levels and

**G** serum creatinine levels of the indicated genotypes of mice at day 15. The following numbers were analyzed: [control=11, STK = 7]. Data presented as median, p-values by two-tailed, Mann-Whitney test. **H** Indirect immunofluorescence for LTL (green) and DBA (red) in the indicated genotypes of mice on post-natal day 15. Scale bar = 100 μm. **I** Low (top) and high (bottom) magnification images of H&E-stained kidneys of the indicated genotypes. Arrow indicates intratubular psammomatous calcification. Scale bar = 100 μm. Representative immunohistochemistry (IHC) for **J** TFE3 (top row), GPNMB (bottom row) **K** Melan A (top row), and PMEL (bottom row) in the indicated genotypes of mice on post-natal day 15. Scale bar = 100 μm. **L** Immunoblotting of kidney lysates from control and *SFPQ-TFE3^LSL; Ksp-Cre* transgenic mice at post-natal day 15 for the indicated antibodies. All experiments represent n ≥ 3 independent biological replicates. Source data are provided as a Source data file.

## Conditional post-natal, doxycycline-mediated induction of SFPQ-TFE3 in Pax8-CreERT mice induces renal tumor development

To examine effects of SFPQ-TFE3 expression in renal tubular cells following completion of kidney development, we leveraged a conditional, tamoxifen-inducible Pax8 Cre-ERT model[33]. Pax8 is a critical renal lineage transcription factor, thus Cre is diffusely expressed in most renal epithelial cells of the proximal and distal renal tubules, loops of Henle, collecting ducts, and the parietal epithelial cells of Bowman's capsule following tamoxifen exposure in this model. *SFPQ-TFE3^LSL; Pax8-CreERT* (*STP*) mice were born at expected Mendelian ratios and survived to adulthood, when they were injected with tamoxifen at 8–12 weeks of age to induce Cre expression. When mice were sacrificed at 3.5 months following tamoxifen, *STP* kidneys were variably enlarged with bilateral solid tumors (Fig. 2A), showing strong, diffuse, nuclear TFE3 induction (Fig. 2B). Tumor cells were diffusely infiltrative around normal renal tubules, leading to disappearance of the normal cortico-medullary junction, and tumor replacement of a large part of the kidney parenchyma (Fig. 2B). Kidney/ body weight ratios and BUN levels were also elevated in some *STP* mice by 3.5 months after tamoxifen, though some variability in tumor burden was evident in this inducible model, with tamoxifen-injected, female *STP* mice frequently demonstrating higher kidney/ body weight ratios and BUN levels than males at 4.5 months, consistent with previous studies in humans and mice[26,34] (Fig. 2C, D). To validate these observed sex-based differences in tumor burden, we digitally annotated histologic tumor volume based on GPNMB positivity in larger cohorts of *STP* mice at 3.5 months, demonstrating statistically significant sex-based differences at this age (Supplementary Fig. 3A, B). Histologically, the tumors were comprised of nests of monomorphic epithelioid cells with clear to eosinophilic cytoplasm and large round nuclei (Fig. 2E). Though they were invasive, the tumor cells were similar in morphology to intratubular proliferations seen in *STK* kidneys (Fig. 1I) and broadly resembled those seen in human PEComas (Supplementary Fig. 3C). By IHC and immunoblotting, *STP* tumor cells showed strong expression of nuclear TFE3, the canonical TFE3 target GPNMB, as well as expression of melanocytic and lysosomal markers (PMEL, MelanA and Cathepsin K) commonly upregulated in human MiT/TFE-related neoplasms (Fig. 2F, G). In contrast, *STP* tumor cells entirely lacked expression of proximal (LTL) and distal (DBA) renal tubular markers (Supplementary Fig. 3D), as well as cytokeratins, such as type II (CK8) or type I keratins (including CK10, 13, 14, 15, 16, 17 and 18) (Fig. 2H, left panels). Notably, *STP* tumor cells showed minimal vimentin expression, and were negative for smooth muscle marker (α-SMA) by IHC (Fig. 2H, right panels). Cumulatively, these findings suggested that post-natal expression of *SFPQ-TFE3* within murine renal tubular cells results in highly penetrant, infiltrative epithelioid tumors, with brisk expression of melanocytic/lysosomal markers. Importantly, these markers are characteristic of human MiT/TFE-related tumors, such as the tRCC cases inadvertently included in The Cancer Genome Atlas (TCGA) papillary

RCC (KIRP) cohort[35] (Fig. 2I) or tuberous sclerosis (TSC)-related PEComas (angiomyolipomas)[3,36] (Fig. 2J).

## mTORC1 signaling is activated in murine and human models of SFPQ-TFE3 fusion-RCC

mTOR signaling has previously been shown to be activated in human tRCC samples and pre-clinical models by transcriptomic and proteomic profiling[18,24,25,27,28], including in a recent mouse model of *ASPSCR1*-TFE3 tRCC[26], though this signaling pathway was not examined in the previously published *PRCC-TFE3* transgenic mice[4]. We performed gene set enrichment analyses (GSEA)[37], using Hallmark gene sets[38] on publicly available gene expression data from the *PRCC-TFE3* mice[4], and found significant enrichment of genes associated with PI3K/ AKT/ MTOR signaling in the transgenic mice at 7 months compared to controls (Supplementary Fig. 4A). By immunoblotting, there were increased levels of mTORC1 substrate p-4E-BP1 in *PRCC-TFE3; KSP-Cre* kidney tumor lysates compared to controls, though total 4E-BP1 levels were increased as well (Supplementary Fig. 4B) as has been seen previously in other mouse models with mTORC1 activation in the kidney[20]. We then characterized mTORC1 signaling in the *SFPQ-TFE3* transgenic models. Phosphorylation of canonical mTORC1 substrates and downstream signaling intermediates (p70 S6K, 4E-BP1 and S6) was elevated in P15 *STK* transgenic kidney tumor lysates compared to controls, by immunoblotting (Fig. 3A) and IHC (Fig. 3B). Levels of phosphorylated canonical mTORC1 substrates and downstream signaling intermediates were also elevated in kidney tumor lysates from 3.5-month *STP* mice compared to treated littermate controls (with substantial increase in total substrate levels seen in these tumors) by immunoblotting (Fig. 3C), and similar evidence of mTORC1 activation seen by IHC (Supplementary Fig. 4C). Notably, in the *STP* mice, expression of phosphorylated mTORC1 substrates were specifically increased in the TFE3 (+) tumor cells by IHC, with low expression in surrounding normal kidney (Fig. 3D). In addition, levels of nuclear TFEB, an established non-canonical mTORC1 substrate[39], were also lower in the tubular cells from STK transgenic mice compared to littermate control kidneys by IHC, and a similar pattern was seen in tumor cells from *STP* transgenic mice, compared to internal control surrounding normal renal tubules (Fig. 3E). These findings are consistent with TFEB hyperphosphorylation and cytoplasmic retention due to increased mTOR signaling.

To validate these findings, we also examined mTORC1 activation in multiple tRCC human cell line models[4], comparing them to ccRCC cell lines by immunoblotting. Phosphorylation of canonical (p70 S6K) and non-canonical (TFEB) mTORC1 substrates and downstream signaling intermediates (S6) was generally elevated in the tRCC cell lines with *SFPQ-TFE3* (UOK145), *PRCC-TFE3* (UOK120, UOK124, UOK146) and *NONO-TFE3* (UOK109) fusion expression, compared to ccRCC controls (UOK111, UOK140 and UOK150) (Fig. 3F). Because comparison of genetically disparate UOK cell lines does not allow examination of the

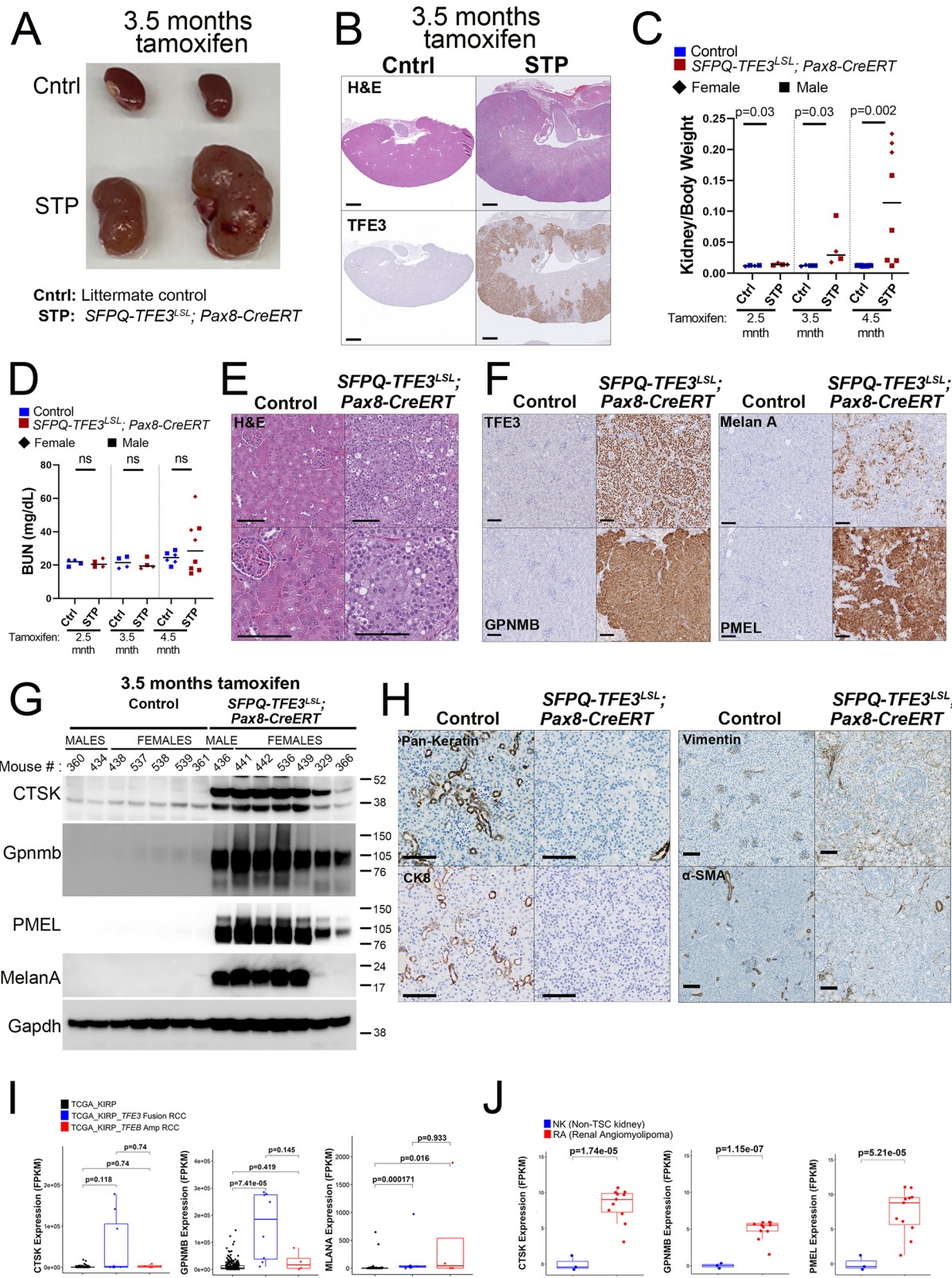

isolated effects of *TFE3* fusions on mTORC1 activation in an isogenic setting, we engineered HEK293 cells with doxycycline-inducible expression of *WT-TFE3*, *S321A-TFE3* (an mTORC1-site phosphodeficient TFE3 point mutant that is constitutively nuclear localized[40]), *SFPQ-TFE3*, *PRCC-TFE3* and *NONO-TFE3*, using the Flp-In-T-Rex™ system in HEK293 cells. Doxycycline-mediated induction of the variably sized fusion proteins and canonical E-box targets *GPNMB*[4], *RRAGD*[41]

and *FLCN*[41] was confirmed by immunoblotting (Supplementary Fig. 4D). Phosphorylation of mTORC1 substrates and downstream signaling intermediates was strongly elevated upon induction of *SFPQ-TFE3* and *PRCC-TFE3* in this system, with subtler induction due to *WT-TFE3*, *S321A-TFE3*, and minimal induction with *NONO-TFE3* (Supplementary Fig. 4D). To validate these results in a renal tubular cell line, we leveraged a previously described HK2 cell system (human proximal

**Fig. 2 | Conditional post-natal, doxycycline-mediated induction of SFPQ-TFE3 in Pax8 Cre-ERT mice induces renal tumor development. A** Macroscopic kidney images from representative *SFPQ-TFE3<sup>LSL</sup>; Pax8-CreERT (STP)* transgenic mice and age-matched, littermate controls, at 3.5 months following tamoxifen. **B** H&E (top-row) and TFE3 IHC (bottom-row) from representative *STP*/control mice, at 3.5 months following tamoxifen. Scale bar: 1 mm. **C** Kidney/body weight ratios, and **D** BUN levels from *STP*/control mice, at 2.5, 3.5 or 4.5 months, following tamoxifen. Kidney/body weight ratios: 2.5 months [control = 4/STP = 4], 3.5 months [control = 4/STP = 4] and 4.5 months [control=12/STP = 8]. BUN levels: 2.5 months [control=4/STP = 4)], 3.5 months [control = 4/STP = 4] and 4.5 months [control=6/STP = 8]. Squares=males, diamonds = females. Data are presented as median. P-values by two-tailed, Mann-Whitney test. **E** Low (top-row) and high (bottom-row) magnification images of kidney H&Es from tamoxifen-injected, *STP*/control mice. Scale bar = 100 μm. Representative IHC for **F** TFE3 (top-row), GPNMB (bottom-row) [left panels], and Melan A (top-row), and PMEL (bottom-row) [right panels], from tamoxifen-injected, *STP*/control mice. Scale bar= 100 μm. **G** Immunoblotting of kidney lysates from tamoxifen-injected, *STP*/

control mice at 3.5 months following tamoxifen. **H** Representative IHC for Pan-Keratin, CK8 (left panels), Vimentin and α-SMA (right panels) from tamoxifen-injected, *STP*/control mice. Scale bar = 100 μm. **I** The TCGA gene expression database for papillary RCC (KIRP) was utilized to compare *Cathepsin K (CTSK)*, *GPNMB* and *MLANA* gene expression in *TFE3* fusion-RCC (n = 8) and *TFEB*-amplified RCC (n = 4) to the remainder of papillary RCC cases without *TFE3* fusions or *TFEB* amplifications (n = 273). *P*-values by Wilcoxon rank sum test adjusted with multiple comparisons using false discovery rate (FDR) method. **J** Comparison of *CTSK*, *GPNMB* and *PMEL* expression (FPKM) in RNA seq data from a panel of non-TSC normal kidneys (n = 3) and renal angiomyolipomas with *TSC1/2* biallelic loss (n = 11)[36]. *P*-values by Wilcoxon rank sum test. For (**I, J**): Two-sided tests were used in the analyses. Data are represented as box-and-whisker plots, where center line indicates median, box edges represent 25th and 75th percentiles, and whiskers extend to the most extreme data points within 1.5× the interquartile range (IQR). Outliers beyond this range are shown as individual points. All experiments represent n ≥ 3 independent biological replicates. Source data are provided as a Source data file.

---

renal tubular epithelial cell line) with doxycycline-inducible expression of *WT-TFE3* and *TFE3* fusion proteins using the rtTA3 (Tet-on) construct[42]. These immunoblotting experiments confirmed increased phosphorylation of mTORC1 substrates and downstream signaling intermediates with expression of *TFE3* fusion proteins in human renal tubular epithelial cells (Supplementary Fig. 4E). A dose response time course for doxycycline confirmed increased TFEB phosphorylation in HK2 cells with inducible SFPQ-TFE3 expression (Supplementary Fig. 4F). Taken together, these data suggest that MiT/TFE fusion gene expression induces increased mTORC1 activity in human cells.

The mechanism of mTORC1 activation downstream of TFEB expression has previously been ascribed to increased MiT/TFE-mediated transcription of *RRAGD*, an essential component of the Rag GTPase nutrient-sensing complex that regulates amino acid-induced mTORC1 activation at the lysosome[19,20,29,30,43]. However, to our knowledge, increased *RRAGD* transcription has been documented in only a single cell line model of tRCC[29]. To explore a potential role for *RRAGD*, or its functionally redundant paralog *RRAGC*[44], in mediating mTORC1 activation in tRCC models, we first examined *RRAGC* and *RRAGD* gene expression in human tRCC specimens[35] (Supplementary Fig. 5A), where both genes were significantly upregulated in tRCC samples compared to papillary RCC without gene fusion expression. We then examined *RRAGC* and *RRAGD* gene expression in the aforementioned panel of 8 UOK tRCC cell lines, and confirmed increased expression of both genes, most notably *RRAGD*, compared to clear cell RCC control cell lines (Supplementary Fig. 5B, C), with similar findings at the protein level by immunoblot (Supplementary Fig. 5D). RagD protein expression was similarly upregulated in HEK293 and HK2 cells upon doxycycline-induced expression of *TFE3* fusion proteins (Supplementary Fig. 4D, E). Finally, we examined *Rragc/d* gene expression in *STK* and *STP* transgenic mice kidneys, where it was also increased compared to control kidneys (Supplementary Fig. 5E, F).

We next examined whether suppression of RRAGC or RRAGD activation via amino acid deprivation or transient *RRAGC* or *RRAGD* silencing was sufficient to suppress mTORC1 signaling in cell line models expressing *TFE3* fusion proteins. In HK-2 cells with inducible *SFPQ-TFE3* expression, amino acid starvation decreased phosphorylation of direct mTORC1 substrates 4E-BP1, p70S6K, and TFEB to levels near control cells lacking fusion induction, though relative increased phosphorylation of substrates in cells expressing *SFPQ-TFE3* compared to those without induction remained evident (Supplementary Fig. 5G). Either transient *RRAGC* and/or *RRAGD* knockdown in *SFPQ-TFE3*-expressing cells resulted in variably decreased phosphorylation of 4EB-P1, S6, and/or TFEB compared to control, though similar results were not seen for p70S6K phosphorylation (Supplementary Fig. 5H). Similarly, we transiently overexpressed increasing concentrations of HA-tagged, inactive RRAGC<sup>GTP</sup> (Q120L) or RRAGD<sup>GTP</sup> (Q121L)[43] in HK2/ *SFPQ-TFE3* cells

(Supplementary Fig. 6A). In cells without doxycycline induction of *SFPQ-TFE3*, over-expression of either inactive RRAGC or RRAGD decreased TFEB phosphorylation as previously described[44,45], while the effects were more modest in cells expressing *SFPQ-TFE3*, with no change in p-TFEB, and a slight reduction of p-4E-BP1 observed at the highest concentrations of inactive RRAGD transfection. Collectively, these data indicate that while RRAGC and RRAGD expression is increased with *TFE3* fusion expression, transient knockdown or expression of inactive mutants of *RRAGC* and/or *RRAGD* is insufficient to completely suppress phosphorylation of mTORC1 substrates in this context.

Because mTORC1 recruitment to the lysosome is mediated by RRAGC/D in wild type cells and is associated with kinase activation, we next examined lysosomal levels of the mTORC1 subunit Raptor in HK-2/ *SFPQ-TFE3* cells. Relative to cytoplasmic levels, lysosomal levels of Raptor were increased in HK-2/ *SFPQ-TFE3* cells with increasing duration of doxycycline (Supplementary Fig. 6B, top panels; Supplementary Fig. 6C, D). As a control, mTORC1 inhibition with Torin1 increased relative lysosomal Raptor as expected[19,46]. In contrast, *RRAGC* or *RRAGD* siRNA only variably and non-significantly decreased lysosomal Raptor levels (Supplementary Fig. 6C, D), consistent with variable effects of *RRAGC* and/or *RRAGD* knock-down on mTORC1 substrate phosphorylation. Taken together, our results suggest that RRAGC and/or RRAGD may contribute to mTORC1 signaling in tRCC but are likely not the only mechanism leading to increased mTORC1 activity.

In addition to *RRAGC/D*, the vacuolar H<sup>+</sup>-ATPse (v-ATPase) is an MiT/TFE transcriptional target[40] and an important component of the lysosomal machinery that activates mTORC1[47]. Reciprocally, mTORC1 activation itself drives v-ATPase expression in cells and mice[48], forming a positive feedback loop. Accordingly, we found that expressions of multiple v-ATPase subunits were significantly elevated in *TFE3*-fusion RCC cases in the TCGA KIRP cohort, compared to ccRCC (Fig. 4A). In HK-2 cells with inducible *SFPQ-TFE3* expression, V-ATPase subunit expression was similarly elevated by qRT-PCR, whole cell- and lyso-somal fractionation- immunoblotting, and down-regulated by mTOR inhibition via Torin1 or *RHEB* siRNA (Fig. 4B-D, Supplementary Fig. 6B-lower panels). Expression of multiple v-ATPase subunits was also variably elevated in the UOK cells with *TFE3*-fusions, compared to ccRCC (Fig. 4E). ATP6V0C is an evolutionarily conserved v-ATPase subunit that interacts with Ragulator and controls mTORC1[47,49]. We found that siRNA-mediated depletion of ATP6V0C was sufficient to decrease p-4EBP1 in HK-2/*SFPQ-TFE3* cells (Fig. 4F). Finally, treatment of HK-2/*SFPQ-TFE3* cells with BafilomycinA1 (BafA1)-a chemical v-ATPase inhibitor that binds to the V0C subunit[50]-downregulated phosphorylation of multiple mTORC1 substrates in a dose- and time-dependent manner (Fig. 4G, Supplementary Fig. 6E), further substantiating the v-ATPase complex as a key contributor to mTORC1 activation in HK-2/*SFPQ-TFE3* cells.

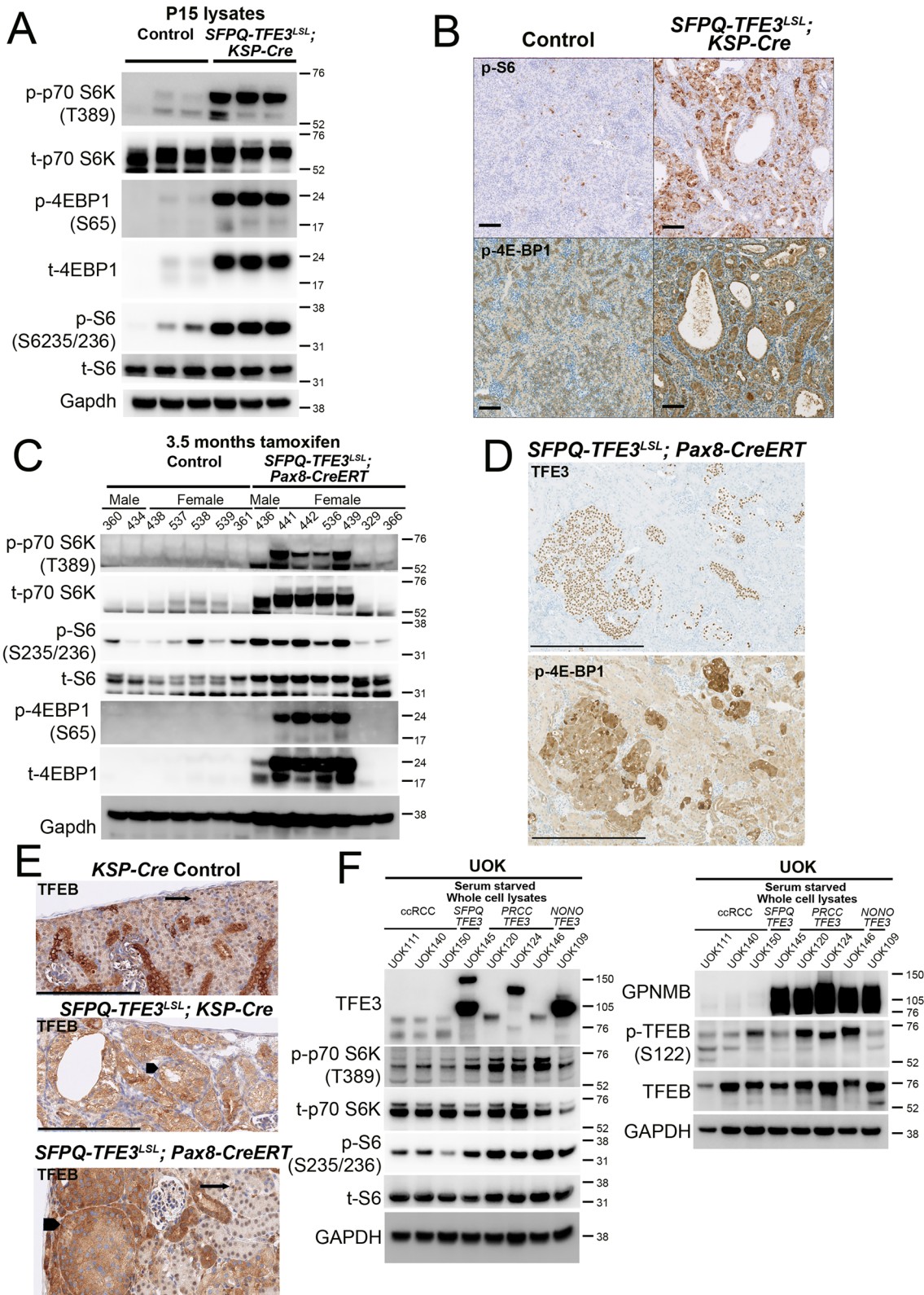

## Induction of SFPQ-TFE3 expression in murine renal tubular epithelial cells results in lineage plasticity with silencing of nephric lineage factors Pax8 and Pax2

Although the tumors in our *STP* model were epithelioid and definitively derived from a PAX8-positive cell of origin due to use of the *Pax8-CreERT* model[33], the lack of keratin and renal tubular marker expression raised the possibility that these tumors more closely

resembled malignant PEComas rather than tRCC. To further characterize the lineage of these tumors, we carried out RNA-seq on 15-day *STK* and 3.5-month tamoxifen-treated, *STP* transgenic kidneys, as well as 7-month *PTK* kidneys, and examined differentially expressed genes compared to their respective normal control kidneys (Supplementary Data 1-3). Notably, the core set of differentially expressed genes overlapping between human tRCC and the *ASPSCR1-TFE3* mouse

**Fig. 3 | mTORC1 signaling is activated in murine and human models of SFPQ-TFE3 fusion-RCC. A** Immunoblotting of kidney lysates from control and *SFPQ-TFE3^{LSL}; Ksp-Cre* transgenic mice at post-natal day 15 for phosphorylation of mTORC1 substrates (p-p70S6K [T389], p-4E-BP1[S65] and p-S6[S235/236]). **B** Representative immunohistochemistry (IHC) for p-S6[S235/236] (top row) and p-4E-BP1[T37/46] (bottom row) from age-matched, control and *SFPQ-TFE3^{LSL}; Ksp-Cre* transgenic mice at post-natal day 15. Scale bar = 100 μm. **C** Immunoblotting of kidney lysates from *SFPQ-TFE3^{LSL}; Pax8-CreERT* transgenic mice and age-matched, littermate controls, at 3.5 months following injection of tamoxifen, for phosphorylation of mTORC1 substrates (p-p70S6K [T389], p-4E-BP1[S65] and p-S6[S235/236]). **D** Representative immunohistochemistry (IHC) for TFE3 (top row) and p-4E-BP1[T37/46] (bottom row) from tamoxifen-injected, *SFPQ-TFE3^{LSL}; Pax8-CreERT*

transgenic mice and age-matched, littermate controls, at 2.5 months following injection of tamoxifen. Scale bar = 500 μM. **E** Representative immunohistochemistry (IHC) for TFEB from *Ksp-Cre* control mice (top row) and *SFPQ-TFE3^{LSL}; Ksp-Cre* transgenic mice (middle row) at post-natal day 15, and from tamoxifen-injected, *SFPQ-TFE3^{LSL}; Pax8-CreERT* transgenic mice at 3 months following injection of tamoxifen (bottom row). Arrows indicate nuclear TFEB in normal tubules. Arrowheads indicate cytosolic TFEB in expanded tubules. Scale bar = 200 μM. **F** Immunoblotting of lysates from *TFE3* fusion-RCC cell lines [UOK145(*SFPQ-TFE3*), UOK120,124,146 (*PRCC-TFE3*) and UOK109(*NONO-TFE3*)], and ccRCC controls (UOK111, UOK140 and UOK150) for phosphorylation of mTORC1 substrates. All experiments represent n ≥ 3 independent biological replicates. Source data are provided as a Source data file.

model[26] was strikingly enriched in the *STP, STK,* and *PTK* transgenic kidney tumors (Supplementary Fig. 7A), highlighting the transcriptional overlap between our mouse models and tRCC. However, gene set enrichment analyses using standard KEGG sets were notable for convergent negative enrichment for the peroxisome and numerous peroxisome-associated metabolic pathways[51] associated with renal tubular epithelial cells in the *STK, STP,* and *PTK* models (Supplementary Data 4–6). Consistent with this finding, there was negative enrichment of genes associated with renal epithelial cell subsets from the cell type signature gene sets (C8) in *STK* and *STP*, and to a lesser extent in *PTK* transgenic kidneys, consistent with downregulation of keratins and renal tubular markers in these models (Supplementary Fig. 7B).

To further probe loss of renal tubular cell identity in the *STP* model, we examined expression of core renal lineage transcription factors in tubular cells with or without *SFPQ-TFE3* expression within the same kidney. We used GPNMB expression to identify renal tubules with mosaic *SFPQ-TFE3* fusion transgene expression in mice treated for only two weeks with tamoxifen. Strikingly, nuclear PAX8 expression was conspicuously absent in GPNMB+ tubular epithelial cells, in contrast to adjacent GPNMB- neighboring tubular cells (Fig. 5A), indicating that suppression of PAX8 occurs rapidly after *SFPQ-TFE3* fusion expression. Concordant results were seen for keratin (CK8) immunostaining at this timepoint, with CK8 loss occurring specifically in cells with fusion gene expression (Supplementary Fig. 8A). After 3.5 months of tamoxifen treatment, the full-blown tumors in the *STP* model lacked detectable PAX8 (Fig. 5B), consistent with loss of keratin in this model (Fig. 2H). A more heterogeneous pattern of PAX8 and PAX2 suppression was seen in the *STK* kidneys at P15 (Fig. 5C) as well as the *PTK* kidney tumors at 7 months age (Fig. 5D) however these differences were highly significant on digital quantification for PAX8 IHC expression (Fig. 5E, F), indicating that both *SFPQ-TFE3* and *PRCC-TFE3* expression drives varying degrees of renal lineage factor loss in the murine kidney.

The *PAX8* transcription factor hub drives a core regulon of target genes in the kidney to promote proximal tubular epithelial cell fate, including *GATA3, LHX1* and *WT1*, among others[52], and we confirmed these targets in HK2 proximal tubular epithelial cells following *PAX8* knockdown using immunoblotting and qRT-PCR (Supplementary Fig. 8B, C). Loss of PAX8 phenocopied expression patterns in renal TSC-related PEComas (angiomyolipomas), which show striking downregulation of *PAX8* or *PAX2* and their core target genes (*GATA3, WT1, LHX1*)[36,52] compared to surrounding kidney parenchyma. (Supplementary Fig. 8D). Indeed, expression of GATA3 and pan (type I) keratin was also markedly decreased in *STP* tumors by 3.5 months after tamoxifen treatment, consistent with loss of PAX8 (Supplementary Fig. 8E) and *STK, STP* and *PTK* transgenic kidneys showed significant enrichment of both up- and downregulated genes from renal TSC-related PEComas compared to controls (Fig. 5G)[36]. Finally, to explore the temporal dynamics of downregulation of *PAX8, PAX2* and downstream target genes following *SFPQ-TFE3* fusion expression, we leveraged cultured primary renal tubular epithelial cells from *ST* mice,

treated with empty or Cre-recombinase-expressing adenovirus in vitro. By immunoblotting, expressions of PAX8, PAX2 and their transcriptional targets GATA3 and WT1, as well as pan-keratin (Type 1) were decreased within two days of *SFPQ-TFE3* induction (Fig. 5H, Supplementary Fig. 8F). Taken together, both *SFPQ-TFE3* and *PRCC-TFE3* fusion expression are accompanied by downregulation of renal lineage transcription factors and their downstream targets in vivo and in vitro in the murine kidney. This finding is most dramatic in the *STP* model, where resulting tumors are best classified as renal epithelioid malignant PEComas, based on total loss of PAX8, PAX2, cytokeratin and renal tubular marker expression, with accompanying upregulation of melanocytic and lysosomal markers. Since *SFPQ-TFE3* expression is limited to PAX8-expressing cells in the *STP* model due to use of the *Pax8-CreERT* model, this model serves as a lineage tracing experiment, thereby substantiating renal tubular epithelial cells as the cell of origin for a TFE3 fusion-driven murine PEComa model.

## Induction of SFPQ-TFE3 expression in human renal tubular epithelial cells results in lineage plasticity with silencing of nephric lineage factors PAX8 and PAX2

To validate our murine results in human systems and to further test whether findings were conserved across different *TFE3* fusion partners, we next examined expression of PAX8 and PAX2 in the HK2 proximal renal epithelial cell line with doxycycline-inducible expression of common *TFE3* fusions. Similar to findings in our mouse models, PAX8 and PAX2 protein expression were most strikingly and specifically downregulated with induction of *SFPQ-TFE3* and this was accompanied by robust upregulation of the melanocytic marker PMEL (Fig. 6A). Notably, WT-TFE3 over-expression did not affect PAX8 or PAX2 expression, while *PRCC-TFE3* induction was accompanied by a mild suppression of PAX2, without discernable effects on PAX8 expression. PAX8 nuclear localization was also significantly decreased in HK2/*SFPQ-TFE3* cells by nuclear-fraction immunoblotting (Fig. 6B) and immunofluorescence (Fig. 6C). These findings were corroborated at the gene expression level, where *PMEL* and *GPNMB* expression were dramatically increased upon *SFPQ-TFE3* induction (Fig. 6D), and accompanied by decreased mRNA expression of *PAX8, PAX2* and the downstream transcriptional targets of these renal lineage transcription factors, including *LHX1, HNF1B* and *GATA3* (Fig. 6E). The downregulation of PAX8 with doxycycline induction of *SFPQ-TFE3* was also rescued in HK2/*SFPQ-TFE3* cells stably expressing TFE3-targeting CRISPR sgRNAs (Fig. 6F), providing evidence that PAX8 silencing is due to *SFPQ-TFE3* expression and not a confounding effect of doxycycline treatment in this system.

Though tRCC is distinguished from PEComa by its retention of detectable keratin and PAX8 expression in clinical practice, the downregulated expression of PAX8 expression in the *PTK* tRCC model and the frequent underexpression of cytokeratin and EMA expression seen in human tRCC[18] led us to test whether there is partial *PAX8* and *PAX2* loss in human tRCC samples that had not been previously appreciated. We first examined the series of patient-derived tRCC cell lines, where UOK145 cells expressing *SFPQ-TFE3* showed a striking

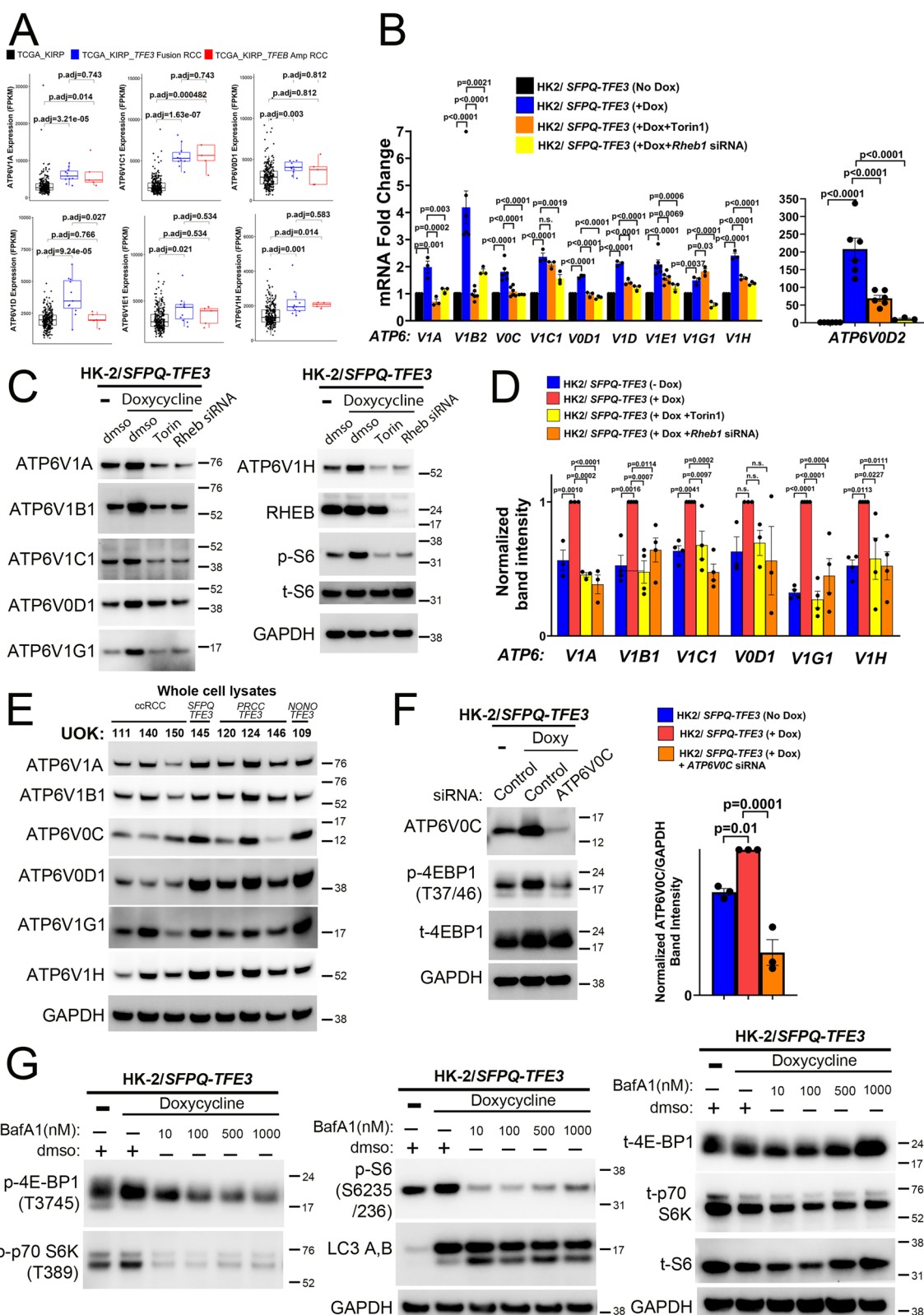

reduction in PAX8 protein expression compared to ccRCC cell lines, but nearly all tRCC cell lines showed some decrease in PAX2 expression compared to the average expression seen in ccRCC lines (Fig. 6G, H). Leveraging the TCGA papillary RCC (KIRP) cohort[35,53], mRNA expression levels of *PAX2* and *PAX8* were significantly decreased in *TFE3* fusion-RCC cases, compared to ccRCC (Fig. 6I), while expression levels of related family members *PAX5* and *PAX6* remained unaltered

(Supplementary Fig. 8G), though sample numbers were too small to examine the effects of specific fusions on *PAX8* and *PAX2* expression. Cumulatively, our results in human samples indicate that *SFPQ-TFE3* expression is associated with particularly potent suppression of renal lineage transcription factor expression compared to other common *TFE3* fusion genes, with resulting lineage plasticity towards a PEComa phenotype, while other fusion genes have a milder effect. These

**Fig. 4 | mTORC1 signaling is activated in murine and human models of SFPQ-TFE3 fusion-RCC. A** Comparison of gene expression for the indicated v-ATPase subunits in *TFE3* fusion-RCC (n = 8) and *TFEB*-amplified RCC (n = 4) to the remainder of papillary RCC cases (n = 273) in the papillary RCC (KIRP) cohort from TCGA. Two-sided tests were used in the analyses. Data are represented as box-and-whisker plots, where the center line indicates the median, the box edges represent the 25th and 75th percentiles, and whiskers extend to the most extreme data points within 1.5× the interquartile range (IQR). *P*-values indicated are by Wilcoxon rank sum test adjusted with multiple comparisons using the false discovery rate (FDR) method. **B** Quantitative real time PCR (qRT-PCR) for the indicated v-ATPase subunits in untreated or doxycycline-treated HK2/*SFPQ-TFE3* cells, following treatment with Torin1 or *RHEB1* siRNA. Replicates: a) V1A, V1C1, V0D1, V1D, V1G1, V1H (n = 3), b) V1B2, V0C, V1E1 and V0D2 (n = 6 for no dox, dox and Torin1 and n = 3 for *RHEB1* siRNA). Graphs represent mean values; error bars represent SEM; *p*-values by one-way ANOVA with Dunnett's test for multiple comparisons. **C** Immunoblotting of untreated/doxycycline-treated HK2/*SFPQ-TFE3* cells, following treatment with Torin1 or *RHEB1* siRNA, for indicated antibodies. **D** Densitometry quantification of normalized protein expression for indicated v-ATPase subunits, in HK2/*SFPQ-TFE3* cells, from experiments in (C). Replicates: a) V1A and V0D1 (n = 3), b) V1B1, V1C1, V1G1 and V1H (n = 4). Graphs represent mean values; error bars represent SEM; *p*-values by one-way ANOVA with Dunnett's test for multiple comparisons. **E** Immunoblotting of lysates from *TFE3* fusion-RCC cell lines and ccRCC controls for indicated antibodies. **F** Immunoblotting (left panel) and densitometry quantification of normalized ATP6V0C protein expression (right panel), from untreated/doxycycline-treated HK2/ *SFPQ-TFE3* cells, following treatment with *ATP6V0C* siRNA (also see Fig. 7I). (n = 3; graphs represent mean values; error bars represent SEM; p values by one-way ANOVA with Dunnett's test for multiple comparisons). **G** Immunoblotting of doxycycline-treated HK2/*SFPQ-TFE3* cells, following treatment with a BafilomycinA1 concentration gradient for indicated antibodies (also see Fig. S9G). All samples are derived from the same experiment- t-4E-BP1, t-p70S6K, t-S6 and GAPDH (right panels) were processed in parallel on a different gel. All experiments represent n ≥ 3 independent biological replicates. Source data are provided as a Source data file.

findings are entirely consistent with the fact that *SFPQ* is the most common *TFE3* fusion partner in human PEComas and further support the fidelity of current MiT/TFE mouse tumor models, where the *PRCC-TFE3* model shows partial retention of PAX8 and epithelial features consistent with tRCC[4], while our *SFPQ-TFE3* model most closely approximates a malignant epithelioid PEComa phenotype.

To investigate potential mechanisms by which SFPQ-TFE3 expression down-regulates *PAX8* mRNA expression, we performed chromatin immunoprecipitation sequencing (ChIP-seq) for *SFPQ-TFE3* in 2 independent clones of HK2/*SFPQ-TFE3* cells. HOMER motif analysis revealed an expected enrichment in CA[C/T] [G/A]TG sequences consistent with the E-box/TFE3-binding motif (Supplementary Data 7–10). Additionally, several de novo motifs (e.g. TGA[G/C] TCA) were also enriched, indicative of potential binding by TFs like Jun, AP1, Fosl2, etc, or RUNX1 as recently described[54]. Notably, we observed strong binding peaks of HA-SFPQ-TFE3 within intragenic and upstream transcription start site regions of the *PAX8* gene (Supplementary Fig. 8H, Supplementary Data 11–12), with identical peaks observed for both clones (e.g. 22831 in PSF6_8_12 and 14370 in PSF6_9_9). ChIP-qPCR using primers specific for the indicated peaks confirmed binding of *SFPQ-TFE3* to these upstream and intragenic regions of *PAX8* following doxycycline treatment in HK2/SFPQ-TFE3 cells (Supplementary Fig. 8I). These SFPQ-TFE3 binding peaks within *PAX8* corresponded to histone modification marks associated with enhancer activity [e.g. H3K27ac binding tracks] in nephron organoids from ENCODE data (Supplementary Fig. 8H), suggesting SFPQ-TFE3 binding at regulatory regions within *PAX8* that could impact its expression.

## mTOR inhibition rescues PAX2 and PAX8 expression and activation in human in vitro models of SFPQ-TFE3 expression

Next, we examined whether mTOR signaling activity is required for renal tubular cell lineage plasticity downstream of *SFPQ-TFE3* fusion expression. Leveraging our in vitro models for pharmacologic and genetic mTOR modulation, we first asked whether suppression of mTOR signaling concurrent with induction of fusion TFE3 expression in HK2/*SFPQ-TFE3* cells affected PAX2/PAX8 expression or MiT/TFE transcriptional target gene expression. Treatment of HK2/*SFPQ-TFE3* cells with the mTOR kinase inhibitor Torin1 rescued PAX2 and PAX8 protein expression in a dose-dependent fashion and simultaneously downregulated expression of the PEComa marker PMEL, as well as multiple MiT/TFE-regulated proteins (LC3A/B, RRAGD, RAB7A, and GPNMB), by immunoblotting (Fig. 7A, B, Supplementary Fig. 9A, B). These effects were reproduced by pre-treatment of HK2/*SFPQ-TFE3* cells with *RHEB* siRNA to knock down a key component of the mTOR complex prior to doxycycline induction (Fig. 7C, D, Supplementary Fig. 9B), providing genetic confirmation. *SFPQ-TFE3*-induced downregulation of keratin expression was also completely rescued by Torin1 and *RHEB* siRNA (Supplementary Fig. 9B). Strikingly, mTOR inhibition concurrent with doxycycline induction via either Torin1 or *RHEB* siRNA was associated with significantly lower *SFPQ-TFE3* fusion protein expression, and this was particularly evident for *RHEB* siRNA (Fig. 7A–D, Supplementary Fig. 9A, B) and mildly dose-dependent for Torin1 treatment (Supplementary Fig. 9A). Accordingly, mTORC1 inhibition with Torin1 rescued PAX8 nuclear localization in doxycycline-treated HK2/*SFPQ-TFE3* cells, by nuclear-fraction immunoblotting (Fig. 7E) and immunofluorescence (Fig. 7F), with both experiments demonstrating a concurrent decrease in nuclear *SFPQ-TFE3* levels. Doxycycline-mediated induction of *SFPQ-TFE3* in HK2 cells resulted in a change in morphology from a cobblestone, epithelial pattern to a more mesenchymal appearance, which was also completely reversed on mTOR inhibition (Supplementary Fig. 9C). The dose-dependent decrease in *TFE3* fusion protein expression upon Torin1 and *RHEB* siRNA treatment was generalizable to other cell line systems and *TFE3* fusion partners examined by immunoblotting, including HEK293 cells with doxycycline-inducible expression of *SFPQ-TFE3* (Supplementary Fig. 9D) or *PRCC-TFE3* (Supplementary Fig. 9E) as well as UOK120 and UOK124 cells (Supplementary Fig. 9F) constitutively expressing *PRCC-TFE3*. At the gene expression level, Torin1 treatment of HK2/*SFPQ-TFE3* cells downregulated MiT/TFE transcriptional targets (GPNMB and PMEL) (Fig. 7G), rescued *PAX2* and *PAX8* gene expression and also upregulated expression of PAX2/8-regulated transcripts (*LHX1, GATA3, HNF1B*), by qRT-PCR (Fig. 7H). These findings are consistent with the fact that PAX2/8 regulation by *SFPQ-TFE3* is mediated at the gene expression level, downstream of elevated mTOR activity.

Importantly, genetic or pharmacological inhibition of the v-ATPase with *ATP6V0C* siRNA (Fig. 7I), or BafilomycinA1 (Supplementary Fig. 9G, H) markedly decreased SFPQ-TFE3 fusion protein expression and rescued PAX8 expression (Fig. 7I), consistent with potent mTORC1 suppression exerted by v-ATPase inhibition (Fig. 4F, G, Supplementary Fig. 6E). In contrast, transient knockdown (Supplementary Fig. 5H), or expression of inactive mutants of *RRAGC* and/or *RRAGD* (Supplementary Fig. 6A), had no effect on SFPQ-TFE3 or PAX8 expression, indicative of a more modest impact of RRAGC/D silencing on mTORC1 signaling in this context. Taken together, these findings suggest that inhibition of v-ATPase expression and/or activity, but not RRAGC/D expression or activity, is sufficient to disrupt the reciprocal SFPQ-TFE3-mTORC1 feedback loop and restore PAX8 expression in the context of inducible *SFPQ-TFE3* expression.

We next examined the mechanisms by which mTORC1 signaling regulates *SFPQ-TFE3* fusion expression. Treatment of HK2/*SFPQ-TFE3* cells with the translation inhibitor cycloheximide revealed similar *SFPQ-TFE3* degradation kinetics in vehicle- or torin1-treated cells, suggesting that the decrease in *SFPQ-TFE3* expression with mTORC1

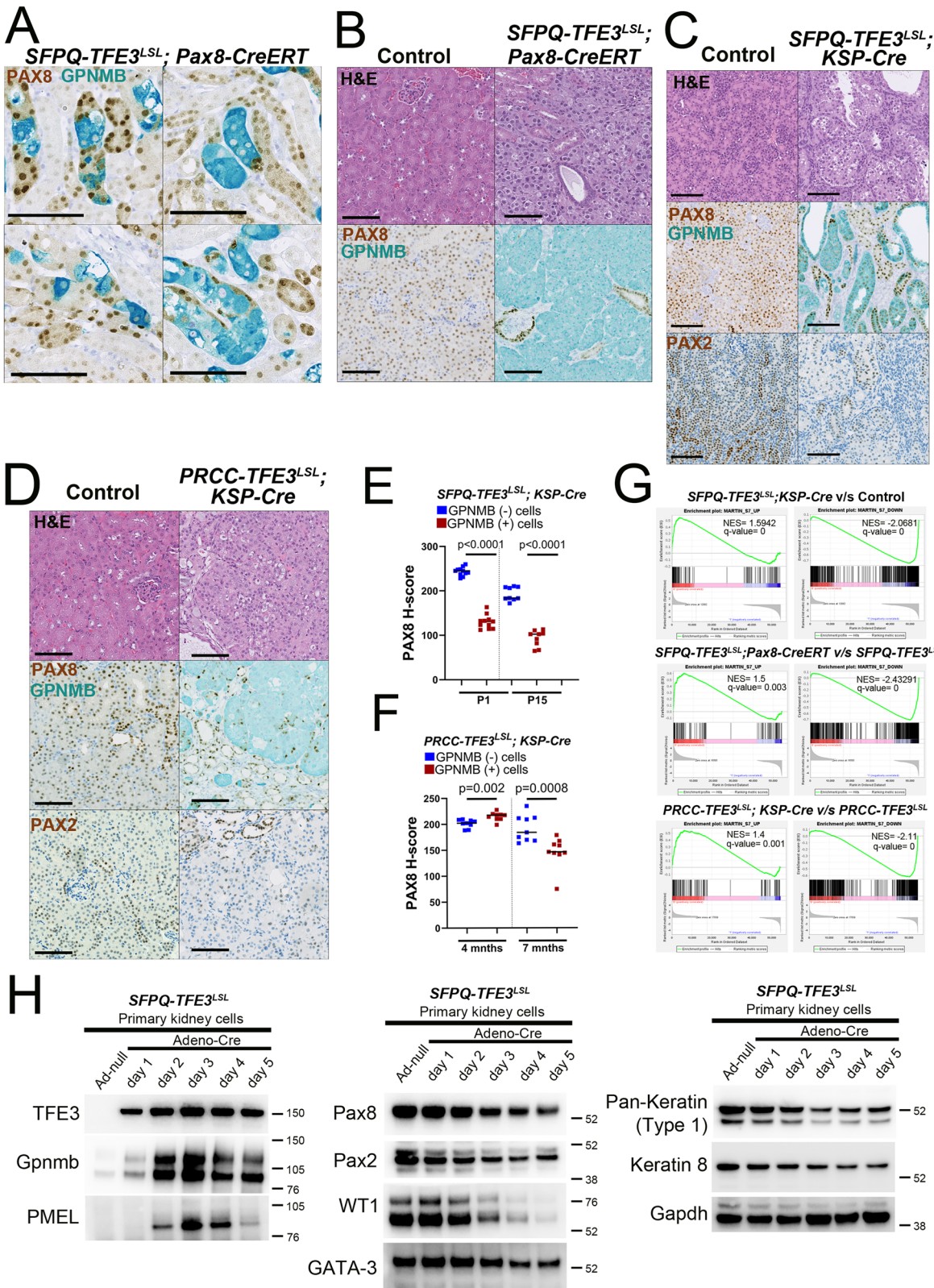

inhibition was not due to decreased protein stability or increased degradation (Supplementary Fig. 10A, B). However, we further investigated this hypothesis in additional experiments. Since mTORC1 inhibition may increase autophagic degradation of SFPQ-TFE3 protein, we also employed a fluorescent spectrophotometric approach to quantitatively estimate autophagic flux or intracellular Cathepsin B activity in vitro. Both CTSB activity (Supplementary Fig. 10C) and Cyto-

ID fluorescence (Supplementary Fig. 10D), were expectedly elevated upon induction of *SFPQ-TFE3*, consistent with increased lipidated LC3-II observed on immunoblotting (Supplementary Fig. 9B). mTOR inhibition with Torin1 or *RHEB* siRNA actually decreased LC3-II (likely due to downregulation of *SFPQ-TFE3*-associated LC3 gene expression/LC3-I protein expression), and neither CTSB activity nor Cyto-ID fluorescence were altered by mTORC1 inhibition. Accordingly, blocking

**Fig. 5 | Induction of SFPQ-TFE3 expression in murine renal tubular epithelial cells results in lineage plasticity with silencing of nephric lineage factors, Pax8 and Pax2. A** Dual IHC for PAX8 (brown) and GPNMB (teal) in tamoxifen-injected, *SFPQ-TFE3^LSL*; *Pax8-CreERT* (STP) transgenic mice and age-matched, littermate controls, at 2 weeks following injection of tamoxifen. Scale bar = 100 μm. **B** H&E (top row) and dual PAX8/GPNMB IHC staining (bottom row) of kidneys from representative tamoxifen-injected, STP mice and controls, at 3.5 months following injection of tamoxifen. Scale bar = 100 μm. **C** H&E (top row), dual PAX8/GPNMB IHC staining (middle row) and PAX2 IHC staining (bottom row) of kidneys from representative age-matched, control and *SFPQ-TFE3^LSL*; *Ksp-Cre* transgenic mice at post-natal day 15. Scale bar = 100 μm. **D** H&E (top row), dual PAX8/GPNMB IHC staining (middle row) and PAX2 IHC staining (bottom row) of kidneys from representative age-matched, control and *PRCC-TFE3^LSL*; *Ksp-Cre* transgenic mice at 7–9 months. **E** Digital quantification of mean nuclear PAX8 H-scores in GPNMB-cells (blue) and GPNMB+ cells (red) at day 1 and day 15, from experiments in **C**, as depicted by scatter plots, with the central line denoting the median. n = 11 (day 1) and n = 9 (day 15) for both, GPNMB − and + cells. *p*-values by two-tailed, Mann-Whitney test. **F** Digital quantification of mean nuclear PAX8 H-scores in GPNMB-cells (blue) and GPNMB+ cells (red) at 4 months and 7 months, from experiments in (**D**), as depicted by scatter plots, with the central line denoting the median. n = 9 at 4 and 7 months, for both, GPNMB − and + cells. *p*-values by two-tailed, Mann-Whitney test. **G** Gene Set Enrichment Analysis (GSEA) comparing 15-day STK (top panels), 3.5-month tamoxifen-treated, STP (middle panels) and 7-month PTK (bottom panels), transgenic kidneys and their controls, for genes differentially expressed in renal TSC-related PEComas from ref. 36. **H** Immunoblotting of lysates from primary renal tubular epithelial cells from *SFPQ-TFE3^LSL* transgenic mice treated with control (harvested at day 5) or Cre-recombinase expressing adeno-virus in vitro for the indicated antibodies (also see Fig. S8F). All experiments represent n ≥ 3 independent biological replicates. Source data are provided as a Source data file.

autophagic flux with concurrent treatment with BafilomycinA1 did not reverse the Torin1-induced decrease in SFPQ-TFE3 protein levels (Supplementary Fig. 10E). Co-immunoprecipitation-ubiquitination analyses of HK2/*SFPQ-TFE3* cell lysates showed a slight decrease in Ub chains associating with immunoprecipitated SFPQ-TFE3 following Torin1 or *RHEB* siRNA treatment, proportionate to the decrease in total SFPQ-TFE3 protein expression with mTORC1 inhibition (Supplementary Fig. 10F, right panels), and concurrent proteasomal inhibition with MG132 or Bortezomib also did not reverse the Torin1-induced decrease in SFPQ-TFE3 protein levels (Supplementary Fig. 10G). Given the lack of evidence for altered fusion protein stability or degradation with mTOR inhibition, we then examined impacts on fusion gene transcription. Significantly, mTORC1 inhibition with Torin1 or *RHEB* siRNA reduced *SFPQ-TFE3* mRNA levels in doxycycline-induced, HK2/*SFPQ-TFE3* cells (Supplementary Fig. 10H) and also decreased *PRCC-TFE3* mRNA expression in UOK124 cells (Supplementary Fig. 10I), by qRT-PCR. Cumulatively, these results suggested that mTORC1 signaling most likely regulates *SFPQ-TFE3* fusion expression via impacts on fusion gene expression levels.

Finally, we examined whether mTOR inhibition via Torin1 reduced kidney tumor burden in *STP* mice. Following induction of *SFPQ-TFE3* expression with tamoxifen, we treated male and female cohorts of mice with vehicle or Torin1 for 4 weeks, following which kidney FFPE sections were immunostained for p-S6 to confirm mTORC1 inhibition (Supplementary Fig. 11A) and GPNMB to quantify tumor area in digital whole slide images of kidney (Supplementary Fig. 11B). While Torin1 treatment did not significantly impact renal tumor burden in female *STP* mice, there was a small and borderline significant reduction in tumor area in male *STP* kidneys, which show a lower tumor burden overall to their female counterparts (Supplementary Fig. 11C). Taken together, our findings indicate that *SFPQ-TFE3* fusion expression is sufficient to induce renal tumorigenesis in mouse models, driving V-ATPase expression and concomitant downstream mTORC1 signaling which, in turn, positively and reciprocally increases *SFPQ-TFE3* gene expression levels. This positive feedback cycle reinforces a lineage switch in renal tubular cells, driving cells towards a PEComa cell phenotype characterized by reduced transcription of renal lineage transcription factors. Inhibition of mTOR signaling interrupts this positive feedback, and by reducing *SFPQ-TFE3* fusion gene expression levels, indirectly rescues PAX8 expression and restores renal lineage identify in vitro but requiring additional work to substantiate in vivo impact (Fig. 8).

## Discussion

tRCC is characterized by minimal somatic alterations[53] other than MiT/TFE-fusion events, which are considered necessary and sufficient to induce renal tumor development. However, there is considerable phenotypic heterogeneity in renal tumors driven by different *TFE3*-fusion proteins, in part due to varying functions of the fusion partners,

underscoring a need to develop fusion-specific models to study the disease. Here, we provide a characterization of constitutive and inducible transgenic mouse models of the *SFPQ-TFE3* fusion oncoprotein-mediated tumorigenesis in the kidney. Similar to previous studies with constitutive expression of *ASPSCR1-TFE3*[26], wild-type TFEB[55], or inactivation of *FLCN*[56,57] during renal development, expression of SFPQ-TFE3 using Ksp-Cadherin-Cre (*SFPQ-TFE3^LSL*; *Ksp-Cre* mice; *STK*) resulted in disrupted renal development and renal insufficiency, culminating in early neonatal death. In contrast, constitutive induction of *PRCC-TFE3* resulted in a cystic renal phenotype that was compatible with life and resulted in delayed tumorigenesis[4]. These variations across constitutive models likely reflect a combination of factors: a) the timing of gene perturbation, since the pathological consequences of gene inactivation are strongly influenced by the developmental status of the kidney, as has previously been well described in the case of *PKD1* deletion[58], b) the tubular origin of the Cre driver used to express the transgene, and perhaps most significantly, c) the fusion gene subtype.

In contrast to constitutive early expression using Ksp-Cadherin-Cre, tamoxifen-induced, postnatal induction of SFPQ-TFE3 [*SFPQ-TFE3^LSL*; *Pax8-CreERT* mice, *STP*], resulted in large, highly penetrant, infiltrative, epithelioid tumors without renal developmental defects. Strikingly, and previously unreported in other fusion-TFE3 models, *SFPQ-TFE3* expression in both constitutive and inducible models was sufficient to downregulate expression of renal lineage transcription factors PAX8 and PAX2 in vitro and in vivo, with evidence of SFPQ-TFE3 binding to PAX8 regulatory regions in vitro. While the *ASPSCR1-TFE3* fusion model resulted in extrarenal PEComas[26], PAX8 was widely expressed in the renal tumors in this model, suggesting that these are best classified as tRCC, though quantification of PAX8 levels relative to normal kidney was not performed. Similarly, the *PRCC-TFE3* model showed retained (though, as we demonstrate, focally low) expression levels of PAX8, consistent with their classification as tRCC[4]. In contrast, postnatal induction of *SFPQ-TFE3* in all PAX8-expressing tubular cells resulted in non-epithelial renal tumors that reproduced the morphology and immunophenotype of human PEComas without PAX8 expression, underscoring the more pronounced role of *SFPQ-TFE3* in driving lineage plasticity, and consistent with the high prevalence of this fusion in human PEComas[3]. Taken together, these models suggest that *TFE* fusions are associated with a continuum of tumor histologies (from tRCC to PEComa), corresponding to a continuum of renal lineage factor downregulation, with *SFPQ-TFE3* at the furthest extreme. Future work will examine what mechanisms might underline these differences and explore whether clinical treatments are best matched to the underlying fusion oncogene rather than the tumor histology (Supplementary Data 13).

The origin and developmental mechanisms that give rise to human PEComas have remained elusive thus far. Cell line models of *TSC* loss[59] or *TFE3*-fusions[60] are frequently characterized by cellular senescence, and this has hampered the identification of a potential cell

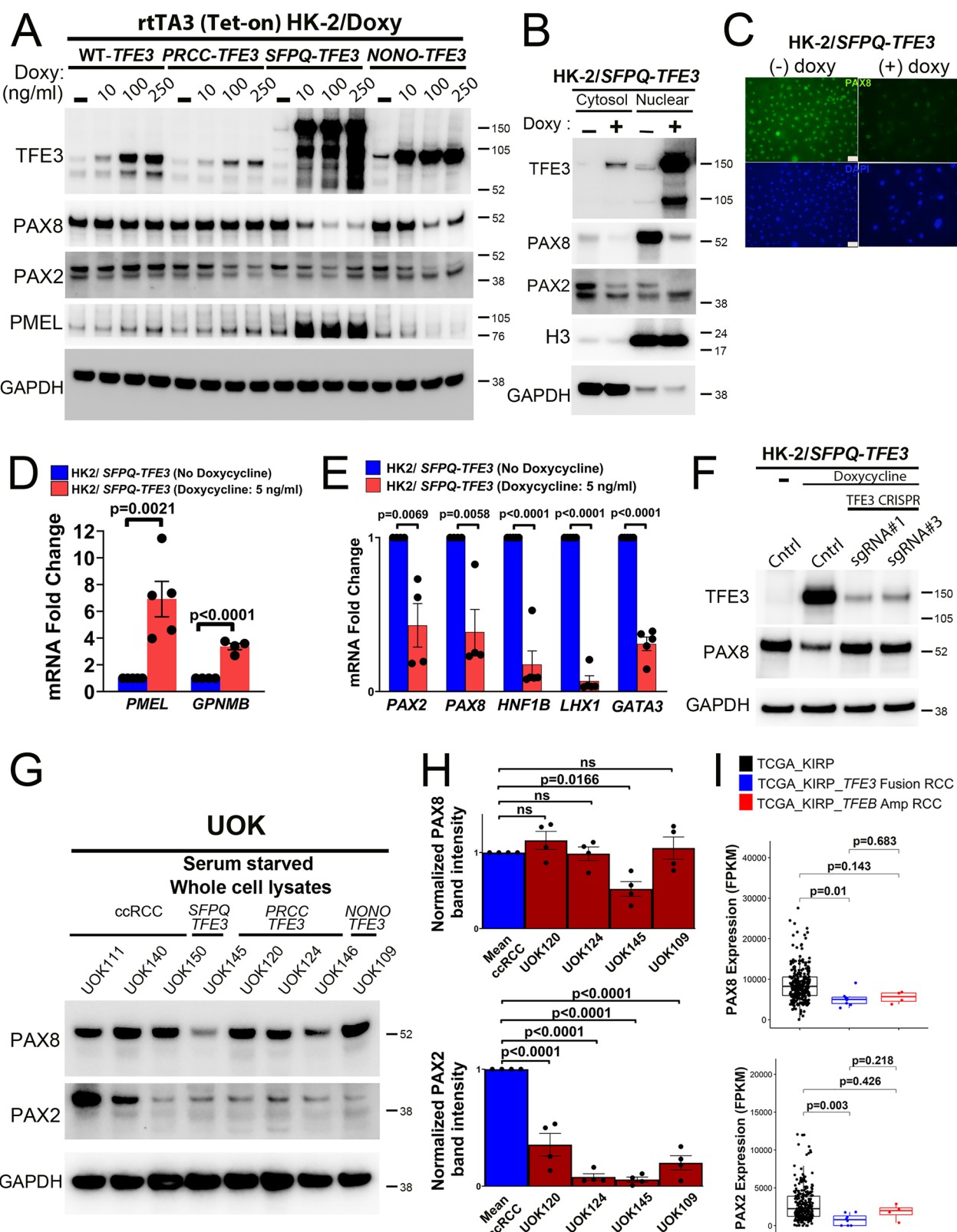

of origin. Renal PEComas are most commonly driven by mTORC1 hyperactivation resulting from biallelic inactivation of *TSC1* or *TSC2*, with both constitutional[15,61] as well as somatic mutagenesis in the *TSC1/2* complex[16,62]. Mechanistically, we and others have shown that mTORC1 activation leads to constitutive activation of *TFEB* and *TFE3*, explaining why TSC1/2 loss and TFE3 fusion expression are mutually exclusive drivers of human PEComas[19,20]. However, homozygous and/or heterozygous deletion of *TSC1* or *TSC2* generally does not induce the formation of renal AML in mice[63,64], although one previous mouse model of mosaic *TSC1* deletion developed renal mesenchymal lesions that resembled human PEComas, with upregulated melanocytic marker expression[65]. Mechanistically, there is evidence that mTORC1 can itself modulate stemness and plasticity[66]: PEComas in TSC have shown resemblance to the embryonic kidney[36] and mTORC1 activation via

**Fig. 6 | Induction of SFPQ-TFE3 expression in human renal tubular epithelial cells results in lineage plasticity with silencing of nephric lineage factors, PAX8 and PAX2. A** Immunoblotting of lysates from HK2 cells with doxycycline-inducible expression of *WT-TFE3* and *TFE3* fusion-proteins, treated with the indicated doses of doxycycline for 72 h. **B** Immunoblotting of cytosolic and nuclear fractions of HK2/*SFPQ-TFE3* cells for indicated antibodies. **C** Indirect immunofluorescence for PAX8 in HK2 cells with doxycycline-inducible expression of *SFPQ-TFE3*. Scale bar = 50 μm. **D** Quantitative real time PCR (qRT-PCR) for *PMEL* and *GPNMB* in HK2 cells with doxycycline-inducible expression of *SFPQ-TFE3*. PMEL (n = 5); GPNMB (n = 4). **E** Quantitative real time PCR (qRT-PCR) for *PAX2, PAX8, HNF1B, LHX1* and *GATA3* in HK2 cells with doxycycline-inducible expression of *SFPQ-TFE3*. PAX2 and PAX8 (n = 4); HNF1B, LHX1 and GATA3 (n = 5). For (**D**) and (**E**), graphs represent mean values; error bars represent SEM; p values by two-tailed, Student's T-test. **F** Immunoblotting of HK2 cells with doxycycline-inducible expression of *SFPQ-TFE3* with CRISPR-Cas9-mediated genomic inactivation of *TFE3*, for TFE3 and PAX8. Two clones representing two unique guide RNAs targeting *TFE3* are shown.

**G** Immunoblotting of lysates from *TFE3*-fusion RCC cell lines [UOK145(SFPQ-TFE3), UOK120,124,146 (PRCC-TFE3) and UOK109(NONO-TFE3)] and ccRCC controls (UOK111, UOK140 and UOK150) for PAX8 and PAX2. **H** Normalized mean densitometric quantifications for PAX8 and PAX2 immunoblots, from experiments in G (n = 4, error bars represent SEM; p values by one-way ANOVA). Mean values from UOK111 and UOK140 lysates were used to calculate mean ccRCC values. **I** Comparison of *PAX8* and *PAX2* gene expression in *TFE3* fusion-RCC (n = 8) and *TFEB*-amplified RCC (n = 4) to the remainder of papillary RCC cases without *TFE3* fusions or *TFEB* amplifications (n = 273) in the TCGA papillary RCC (KIRP) cohort. Two-sided tests were used in the analyses. Data are represented as box-and-whisker plots, where the center line indicates the median, the box edges represent the 25th and 75th percentiles, and whiskers extend to the most extreme data points within 1.5× the interquartile range (IQR). *P*-values indicated are by Wilcoxon rank sum test adjusted with multiple comparisons using the false discovery rate (FDR) method. All experiments represent n ≥ 3 independent biological replicates. Source data are provided as a Source data file.

*TSC1/2* loss in nephron progenitors drives pluripotency along multiple (renal epithelial, stromal, and glial) lineages in the developing kidney[16,61]. mTORC1 activation also induces an EMT program in developing renal organoids which is known to overlap with the stemness transcriptional program, thus offering some insight into their pathogenesis[61,67]. Similar to tumors with *TSC1/2* loss, constitutively nuclear-localized, wild-type TFE3 can also function as a cell-fate switch and prevents lineage commitment and differentiation of embryonic stem cells (ESCs), thereby maintaining a state of pluripotency[68]. Our findings suggest that TFE3 fusion proteins may be the most potent fusion at initiating and maintaining plasticity in the developing and postnatal renal tubular epithelium, with more dramatic and complete trans-differentiation compared to that induced by wild-type TFE3 or *TSC1/2* loss. Future ChIP-seq and ATAC-seq experiments will help to explore how the cistrome landscape and chromatin architecture are uniquely impacted by wild-type TFE3 compared to individual *TFE3*-fusions.

The paired-box transcriptional regulators *PAX2* and *PAX8* are closely related members of the PAX family of genes that are co-expressed in the developing kidney, and are vital for specifying the nephric lineage, survival and morphogenesis during kidney development[69,70]. PAX2 and PAX8 share DNA-recognition motifs and are functionally redundant in several aspects of kidney development such as branching morphogenesis, mesenchymal to epithelial transition and nephron differentiation[71–73]. PAX proteins, via interactions with specific co-factors, are known to act as epigenetic regulators to imprint active or repressive histone marks on chromatin, thus driving renal epithelial cell fate[70]. Accordingly, *Pax2* and *Pax8* double mutant mice exhibit a complete lack of kidney formation, due to an increase in apoptosis during development[71,72]. This developmental reliance on the PAX proteins may in part explain the renal developmental defects in our STK model, where these proteins were dramatically downregulated.

In contrast, the role of the PAX proteins in adult kidneys and renal tumorigenesis is not fully understood. In the adult human kidney, PAX8 is expressed in renal epithelial cells in all nephron segments, including the proximal tubules, renal papillae and in the parietal cells of Bowman's capsule[74], while PAX2 is mainly expressed in the collecting ducts, medulla and renal papillae, where they function to promote water/solute homeostasis and osmotic tolerance[75] or resistance to acute ischemic kidney injury[76,77] in a redundant manner. Increased tumor-associated expression of PAX2 and/or PAX8 is observed in an overwhelming majority of primary and metastatic clear, papillary and chromophobe renal tumors[74,78] and correlates with proliferation index and metastatic disease[79]. Ectopic PAX2 expression is associated with PKD, RCC and Wilms' tumor[80] and PAX8 promotes oncogenic signaling in ccRCC[11,12], suggesting that many renal tumors exhibit a lineage-dependence on PAX2/8 expression to drive tumorigenesis. tRCC retains

expression of PAX8 by definition, however high expression of melanocytic markers and low expression of keratins and EMA corroborates our finding that these tumors downregulate PAX8 relative to the normal kidney[18]. Similarly, emerging evidence from ccRCC suggests, that a subset of the PAX8 regulon comprised of core target genes (*WT1, LHX1, GATA3*) is selectively downregulated with disrupted terminal epithelial differentiation[52]. Furthermore, HNF1B - which is a direct transcriptional target of PAX8[81] - is often silenced in PCKD[82] or chRCC[83], indicating that loss of PAX8 expression or transcriptional activity is compatible with renal tumorigenesis, and may indeed drive lineage plasticity. Intriguingly, PAX2 and PAX8, in addition to specifying nephron identity, also prevent trans-differentiation along alternate lineages: PAX2 loss initiates a cell-fate switch into renal interstitial cells[84], and PAX8 loss impedes the MET process and promotes a mesenchymal phenotype in kidney progenitors[85], consistent with our findings that *SFPQ-TFE3*-induced downregulation of PAX2 and PAX8 in kidney cells may facilitate lineage plasticity and loss of epithelial identity.

Mechanistically, we found that *SFPQ-TFE3* directly bound to regions of the *PAX8* gene associated with active histone modifications like H3K27ac in nephron organoids, and consistent with previous studies showing strong correlation between *TFE3* fusion oncoprotein binding and enhancer formation at genomic loci[54]. Interestingly, SFPQ, by recruiting mSin3A and HDACs[86–88], also functions as a transcriptional co-repressor via gene deacetylation, thus potentially linking *SFPQ-TFE3* binding to observed *PAX8* gene repression. Further studies may indicate whether changes in histone modifications (e.g., H3K27ac, H3K27me3) at the *PAX2/PAX8* loci, or chromatin accessibility changes around *PAX2/PAX8* regulatory regions contribute to SFPQ-TFE3-mediated lineage plasticity.

While studies on the pathogenesis of PEComas have largely established the role of the 2 main genomic drivers (*TSC1/2* loss and *TFE3* fusions), recent work has further characterized their interdependence. *TSC1/2* loss via mTORC1 activation drives *TFEB/TFE3* activity in cellular[48] and mouse models[20] of TSC, via *FLCN* inactivation[19]. Correspondingly, mTORC1 activation, which was hitherto known to be regulated by multiple well-characterized inputs, largely restricted to growth factor signaling (TSC/Rheb complex)[89], nutrient sensing (RAG GTPases)[43] and intra-lysosomal sensing (v-ATPase/ Ragulator complex)[47,90], is now understood to be also regulated by the MiT/TFE factors, with transcriptional induction of *RRAGD* as one of the initial mechanisms described, driving mTORC1 activation in tRCC[19,20,29]. While induction of RagD was more robust compared to RagC in our in vitro models, in HK2/*SFPQ-TFE3* cells, transient silencing, or expression of inactivating mutants of *RRAGC* or *RRAGD* was insufficient to completely suppress mTORC1 activation and consequently failed to downregulate fusion-*TFE3* expression or rescue PAX8 expression. Interestingly, activating mutations in *RRAGD* have been associated with constitutive mTORC1 signaling in kidney

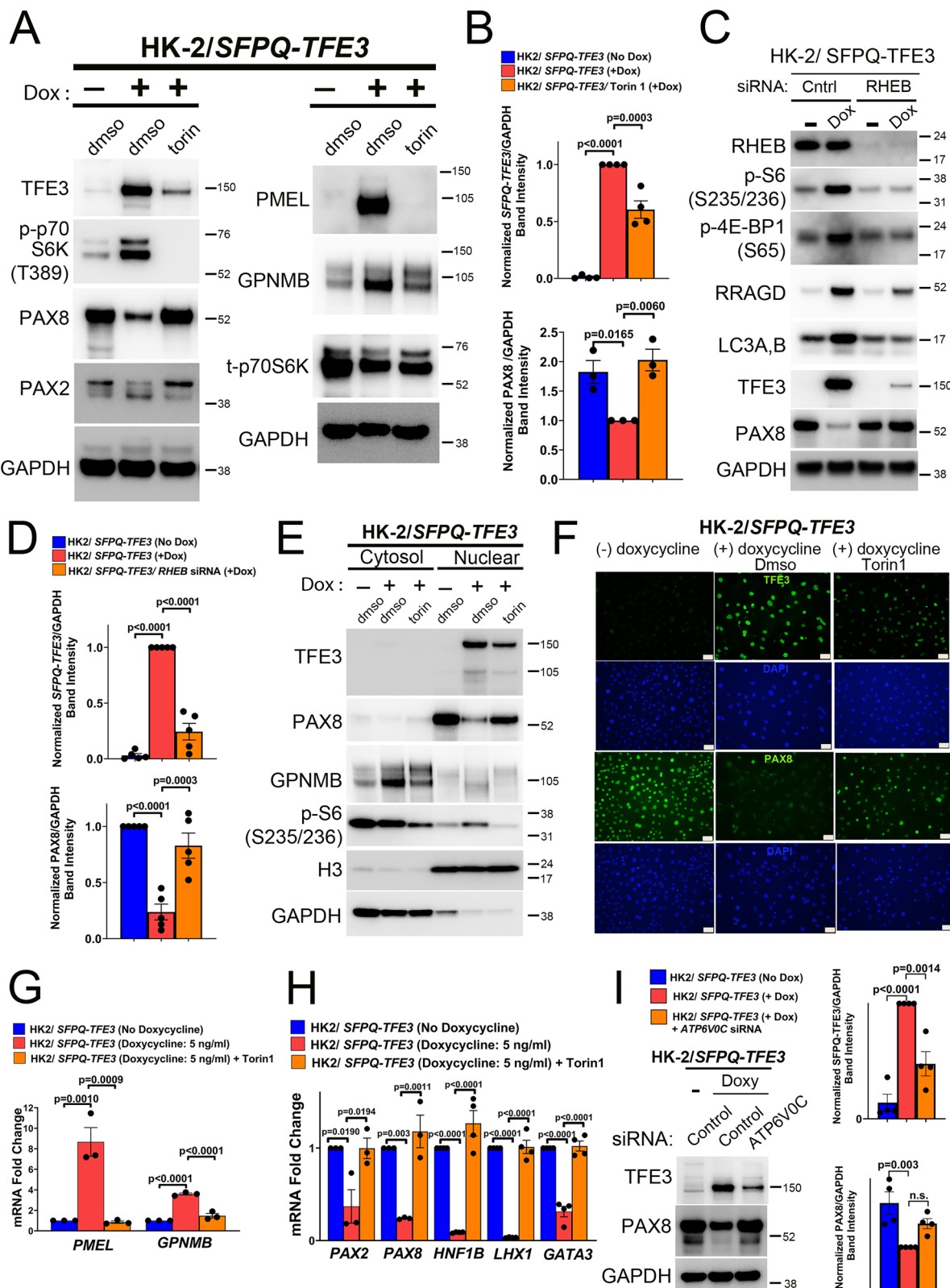

tubulopathies[91,92]. However, RRAGD overexpression alone may be insufficient to regulate mTORC1 activity. Furthermore, in tRCC, in addition to *RRAGD*, multiple regulators of lysosomal mTORC1 signaling (*RHEB*[53], *RRAGB*[26], *RRAGC*[26,53] *FNIP1/2*[26]), are also transcriptionally activated, suggesting a more complex mechanism of mTORC1 activation may be at work. Highlighting this complexity, we now show that expression of multiple V-ATPase components was also

significantly upregulated in *TFE3*-fusion RCC cases within the TCGA, as well as human cell line models of tRCC. Importantly, genetic or pharmacological perturbation of the critical V-ATPase subunit-*ATP6VOC*-was sufficient to downregulate mTORC1 activity, decrease *SFPQ-TFE3* expression and rescue PAX8 expression, highlighting a key role for the V-ATPase complex in the reciprocal regulation of the *SFPQ-TFE3* expression by mTORC1.

**Fig. 7 | mTORC1 inhibition rescues PAX2 and PAX8 expression and activation in human in vitro models of SFPQ-TFE3 expression. A** Immunoblotting of HK2/*SFPQ-TFE3* cells, following treatment with doses of the mTOR kinase inhibitor Torin1, for the indicated antibodies. All samples are derived from the same experiment- t-p70S6K and GAPDH (right panel-bottom 2 blots) were processed in parallel on a different gel. **B** Densitometry quantification of normalized SFPQ-TFE3 and PAX8 expression, in HK2/*SFPQ-TFE3* cells following treatment with Torin1, from experiments in (**A**). SFPQ-TFE3 (n = 4); PAX8 (n = 3). Graphs represent mean values; error bars represent SEM; *p*-values by one-way ANOVA with Dunnett's test for multiple comparisons. **C** Immunoblotting of HK2 cells with doxycycline-inducible expression of *SFPQ-TFE3*, following treatment with *RHEB1* siRNA, for the indicated antibodies. **D** Densitometry quantification of normalized SFPQ-TFE3 and PAX8 expression, in HK2/*SFPQ-TFE3* cells following treatment with *RHEB1* siRNA from experiments in (C). SFPQ-TFE3 (n = 5); PAX8 (n = 5). Graphs represent mean values; error bars represent SEM; *p*-values by one-way ANOVA with Dunnett's test for multiple comparisons. **E** Immunoblotting of cytosolic and nuclear fractions of HK2 cells with doxycycline-inducible expression of *SFPQ-TFE3*, following treatment with

the mTOR kinase inhibitor Torin1 for the indicated antibodies. **F** Indirect immunofluorescence for TFE3 and PAX8 in HK2 cells with doxycycline-inducible expression of *SFPQ-TFE3*, following treatment with the mTOR kinase inhibitor Torin1. Scale bar = 50 µm. **G** Quantitative real time PCR (qRT-PCR) for *PMEL* and *GPNMB* in HK2 cells with doxycycline-inducible expression of *SFPQ-TFE3*, following treatment with the mTOR kinase inhibitor Torin1 (n = 3; error bars represent SEM; *p*-values by one-way ANOVA with Dunnett's test for multiple comparisons). **H** Quantitative real time PCR (qRT-PCR) for *PAX2, PAX8, HNF1B, LHX1* and *GATA3* in HK2 cells with doxycycline-inducible expression of *SFPQ-TFE3*, following treatment with the mTOR kinase inhibitor Torin1 (n = 3; error bars represent SEM; *p*-values by one-way ANOVA with Dunnett's test for multiple comparisons). **I** Immunoblotting (left panels) and densitometry quantification of normalized SFPQ-TFE3 and PAX8 protein expression (right panel), from untreated or doxycycline-treated HK2/*SFPQ-TFE3* cells, following treatment with *ATP6V0C* siRNA. (n = 4, graphs represent mean values, error bars represent SEM, p values by one-way ANOVA with Dunnett's test for multiple comparisons) (also see Fig. 4F). All experiments represent n ≥ 3 independent biological replicates. Source data are provided as a Source data file.

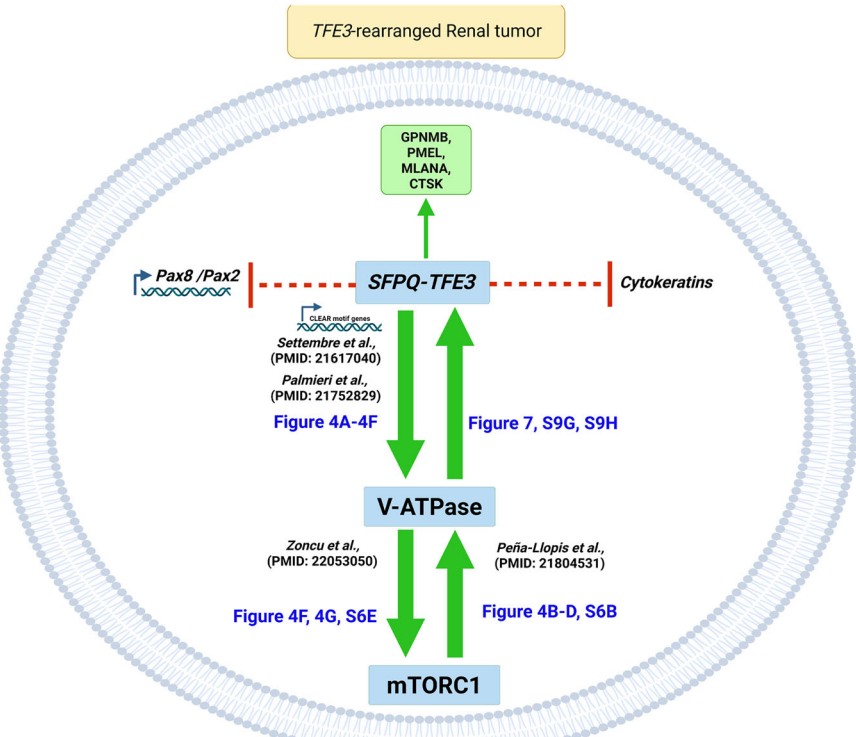

**Fig. 8 | SFPQ-TFE3 reciprocally regulates mTORC1 via the V-ATPase complex.** Model of *SFPQ-TFE3*-mediated mTORC1 activation and tumorigenesis in *TFE3*-rearranged renal tumors. *SFPQ-TFE3* expression is sufficient to induce renal tumorigenesis in mouse models (Fig. 2), and upregulate expression of multiple subunits of the lysosomal V-ATPase complex (Fig. 4A–F), with concomitant activation of downstream mTORC1 signaling (Figs. 3, 4F, G, S4–S6). mTORC1 activation in turn, positively and reciprocally increases *SFPQ-TFE3* gene expression levels via the V-ATPase complex (Figs. 4B–D, 7, S6B, S9G, S9H).This feedback loop

potentiates a lineage switch in renal tubular cells, driving cells towards a PEComa cell phenotype characterized by reduced transcription of renal lineage transcription factors, PAX2 and PAX8 (Figs. 5, 6) and cytokeratin expression (Figs. 2H, S8A, S9B) and upregulated expression of PEComa markers (GPNMB, PMEL, MLANA, CTSK; Fig. 2F, G). Inhibition of mTOR signaling interrupts this positive feedback, and by reducing *SFPQ-TFE3* fusion gene expression levels, indirectly rescues PAX8 expression and restores renal lineage identify in vitro (Fig. 7, S9). Created in BioRender. Asrani, K. (2025) https://BioRender.com/stzijx4.

Underscoring the importance of mTORC1 as an oncogenic driver, multiple groups have reported its activation in pre-clinical models of tRCC. However, studies thus far have demonstrated variable anti-tumor effects of mTOR inhibitors in cell line/xenograft models[27,28,93], transgenic mouse models[26] and in patients for tRCC[94–96] and PEComas[97], with mTOR kinase inhibitors conferring a greater benefit[93] than rapalogs[26] in pre-clinical studies. These findings may potentially be due to robust downregulation of fusion-*TFE3* expression with mTOR kinase inhibitors, as we noted in multiple cell lines expressing *TFE3* fusions. We observed that male STP mice demonstrated a more significant response to Torin1 in vivo, potentially due to a smaller

kidney tumor volume at initiation of the drug, highlighting potential modifying factors in the clinical responses to mTOR kinase therapy in tRCC which should be clarified in further clinical studies.

Unlike wild-type MiT/TFE factors that are inhibited by mTORC1-mediated phosphorylation[98], *TFE3*-fusion proteins have, until now, been considered constitutively nuclear localized and therefore unrestrained by mTORC1[1]. Interestingly, Torin1 induced accelerated degradation of another chimeric oncoprotein (EWS/FLI-1), possibly conferring therapeutic benefit in Ewing's sarcoma[99]. Notably, we found that mTOR inhibition was sufficient to decrease fusion-*TFE3* transcript levels in both inducible, as well patient-derived tRCC cell lines,

consistent with mTOR's previously described role in transcription[100,101]. Future studies may reveal the nature of mTORC1-specific motifs and transcription factor binding elements driving these effects, and additional mechanisms by which *TFE3*-fusions may autoregulate their own transcription in tRCC.

In summary, we describe a transgenic mouse model of renal PEComas induced by *SFPQ-TFE3* expression, the most common *TFE3* gene fusion seen in human PEComas[3,10]. More dramatic than *PRCC-TFE3*, *SFPQ-TFE3* expression is sufficient to rapidly drive lineage plasticity in renal tubular cells, evidenced by renal lineage factor downregulation, loss of cytokeratin expression and melanocytic/lysosomal marker upregulation. These data provide definitive evidence that PEComas may derive from an epithelial cell. Like other fusion-related models of tRCC, *SFPQ-TFE3*-driven renal tumorigenesis is accompanied by early activation of the mTORC1 pathway, mediated by increased V-ATPase levels. We demonstrate that mTOR activation itself positively feeds back to reinforce *TFE3*-fusion gene expression levels, and this feedback is required to drive fusion-related renal lineage factor down-regulation. Future work will further probe the mechanism of epithelial lineage plasticity induced by *TFE3*-fusions and continue to refine the conditions in which mTOR kinase inhibition may have therapeutic efficacy in human fusion-driven tRCC and PEComas.

## Methods

All experiments were performed in compliance with Johns Hopkins University Chemical and Biological Safety and Animal Care and Use Committee regulations.

### Cell culture

UOK cell lines and HK2 cells with stable, doxycycline-inducible expression of TFE3 proteins were a kind gift of Dr. W. Marston Linehan (NCI). HEK293 cells with stable doxycycline-inducible expression of TFE3 proteins were generated using Flp Recombinase-mediated integration, using the Flp-In T-Rex Core Kit (K6500-01, Invitrogen). UOK cells were maintained in DMEM high glucose medium (11995065, Gibco) with l-glutamine, 10% heat-inactivated FBS (SH30071.03HI, Hyclone), 1X MEM (11130051, Gibco) and 1% penicillin/ streptomycin at 37 °C in 5% $CO_2$. Inducible HEK cells were maintained in DMEM high glucose medium, 10% FBS (Tet-approved; A4736401, Gibco), Blasticidin S (15 μg/ml) Hygromycin B (150 μg/ml) and 1% penicillin/streptomycin. Inducible HK-2 cells were maintained in Advanced DMEM/F-12 (12634010, Thermo), 1.5% FBS (Tet-approved) 1X GlutaMAX (35050061, Thermo), Puromycin (0.8 μg/ml), Blasticidin S (2 μg/ml) and 1% penicillin/ streptomycin. For experiments involving amino acid starvation, cells were rinsed in PBS and incubated in amino acid-free DMEM (MBS6120661, MyBioSource) supplemented with 10% dialyzed serum (26400044, Gibco) and glucose (A24940-01, Thermo) for 60−90 min. For amino acid addback, cells were stimulated with a 1X mix of essential (11130036, Thermo Fisher Scientific) and non-essential amino acids (11140035, Thermo Fisher Scientific) and 200 mM L-Glutamine (25030081, Thermo Fisher Scientific), added directly to starved cells for 30 min.

### CRISPR-Cas9 genome editing for TFE3

We designed single-guide RNA (sgRNA) for 3 target sequences in the human *TFE3* gene (GGCGATTCAACATTAACGACAGG, GCGACGCT-CAACTTTGGAGAGGG, TCGCCTGCGACGCTCAACTTTGG) and cloned these into the lentiCRISPR v2 vector (Addgene #52961, Watertown, MA, USA). Lentivirus was produced as previously described[19] and HK-2/ *SFPQ-TFE3* cells were infected for 48 h, selected with puromycin (1 μg/ mL) for 10 days, and colonies established.

### Plasmids, lentiviral transfections and RNAi

Cells were transfected using Lipofectamine 3000 (L3000008, Thermo; plasmid transfections) or Lipofectamine RNAiMAX (13778075, Thermo; siRNA transfections) according to the transfection guidelines. The following siRNAs were used: *RHEB* -siGENOME Human RHEB siRNA SMARTpool (Dharmacon; M-009692-02-0005), *RRAGC*-(Sigma; SASI_Hs01_00190647), *RRAGD*-(Sigma; SASI_Hs01_00057779), *PAX8* (SASI_Hs01_00211299), *ATP6VOC* (SASI_Hs01_00216533).

### Antibodies and reagents

**Primary antibodies.** 4E-BP1 (#9644 CST; 1:2000), CTSK (#ab19027 Abcam; 1:1000), FLCN (#3697 CST; 1:1000), GAPDH (#5174 CST; 1:1000), GATA3(#5852 CST; 1:500), GPNMB (38313 CST; 1:1000), GPNMB (Mouse specific) (#90205 CST; 1:1000), Histone-3 (#4499 CST; 1:4000), HNF4A (#3113 CST; 1:500), Keratin 8 (#ab53280 Abcam; 1:1000), LC3 A, B (#12741 CST; 1:2000), MelanA (#ab210546 Abcam; 1:1000), PAX2 (#9666 CST; 1:500), PAX8 (#59019 CST; 1:500), PMEL (#ab137078 Abcam; 1:1000), Phospho-p70 S6 Kinase (Thr389) (#9205 CST; 1:1000), p70 S6 Kinase (#9202 CST; 1:1000), Phospho-S6 Ribosomal Protein (Ser235/236) (#4858 CST; 1:2000), Phospho-4E BP1 (Ser65) (#9451 CST; 1:2000), Phospho-4E BP1 (Thr37/46) (#2855 CST; 1:1000), p-TFEB(S122) (#86843 CST; 1:2000), p-TFEB(S211) (#37681 CST; 1:2000), Pan-Keratin (Type1) (#83957 CST; 1:1000), RAB7 (#9367 CST; 1:1000), RHEB (#13879 CST; 1:1000), RRAGC (#9480 CST; 1:1000), RRAGD (#4470 CST; 1:1000), Synaptophysin (#5461 CST; 1:1000), S6 Ribosomal Protein (#2317 CST; 1:2000), TFE3 (#14779 CST; 1:4000), TFE3 (#ABE1400 Sigma; 1:4000), TFEB (#4240 CST; 1:2000), WT1 (#83535 CST; 1:1000), Ub (#43124 CST; 1:1000), LAMTOR1 (#8975 CST; 1:1000), RAPTOR (#2280 CST; 1:1000), ATP6V1A (#39517 CST; 1:1000), ATP6V1B1 (#sc-55544 Santa Cruz; 1:500), ATP6V0C (#PA116676 Thermo Fisher; 1:500), ATP6V1C1 (#sc-271077 Santa Cruz; 1:500), ATP6V0D1 (#sc-393322 Santa Cruz; 1:500), ATP6V0D2 (#PA598618 Thermo Fisher; 1:500), ATP6V1G1 (#sc-25333 Santa Cruz; 1:500), ATP6V1H (#sc-166227 Santa Cruz; 1:500), HA-Tag (#3724 CST; 1:1000).

**Reagents.** Tamoxifen (#T5648, Sigma), Torin1 (#14379 CST), Torin1 (#HY-13003, MedChemExpress; for in vivo studies), Doxycycline hyclate (#D9891, Sigma), Cell lysis Buffer (#9803, Cell Signaling), RIPA buffer (#R0278, Sigma), MiniCollect Tube 0.5/0.8 ml CAT Serum Sep Clot Activator (#450533, Greiner), BrdU (#550891, BD Pharmingen), BafilomycinA1 (#54645, Cell Signaling), MG132 (#2194S, Cell Signaling), Bortezomib (#2204S, Cell Signaling), Cycloheximide (#2112, Cell Signaling), Magic Red Cathepsin B kit (#ICT937, BIO-RAD), CYTO-ID (#ENZ-51031, Enzo), Pierce anti-HA-magnetic beads (#88837, Thermo Fisher Scientific), Taqman Gene Expression Master Mix (#4369016, Thermo Fisher), PowerUp SYBR Green Master Mix (#A25742, Thermo Fisher).

### Animal studies.

a) *Animal protocols:* were approved by the JHU Animal Care and Use Committee, under the following protocols: 1) Targeting Lysosomal Biogenesis in Renal Tumors with TSC1/2 Loss (MO23M209) and 2) Targeting GPNMB in renal tumors in tuberous sclerosis complex and translocation renal cell carcinoma (MO22M388).

b) *Animal care conditions:* all animals used had access to food (standard rodent chow) and water ad libitum. Light was regulated by timer, with 12 h on/off cycles. Rooms were maintained at standard mouse temperature (68−79 °F) and humidity (30−70% relative humidity).

c) *Sex as a biological variable*: All studies were performed using both male and female strains of transgenic mice. We observed a higher kidney tumor burden in female tamoxifen-treated, *SFPQ-TFE3*[LSL]; *Pax8-CreERT* mice, consistent with studies in humans, and in previous studies[26,34].

d) *Strains*: The following strains were used: 1) *SFPQ-TFE3*[LSL] mice expressing the *SFPQ-TFE3* fusion downstream of a *LoxP-Stop-LoxP (LSL)* cassette, were generated by Taconic Biosciences. 2) Mice

hemizygous for the _Ksp-Cre recombinase_ knockin gene (Strain Number: 012237) (The Jackson Laboratory). 3) Tamoxifen-inducible, _Pax8-CreERT_ mice were a kind gift of Dr. Athena Matakidou (Cancer Research, UK). _STP_ mice were injected with Tamoxifen at 8-12 weeks of age to induce Cre expression [75 mg tamoxifen/kg body weight via intraperitoneal injection, once every 24 hours for a total of 5 consecutive days]. 4) _PRCC-TFE3^{LSL}_ mice expressing the _PRCC-TFE3_ fusion downstream of a _LoxP-Stop-LoxP (LSL)_ cassette, were a kind gift of Dr. W. Marston Linehan (NCI). Genomic DNA was isolated from tail snips and genotyping performed using the following primers:

1) _SFPQ-TFE3^{LSL}_ - Transgene Forward: 5'-CTT-TAT-TAG-CCA-GAA-GTC-AGA-TGC-3'

   Transgene Reverse: 5'-TGG-AGG-ACA-TTC-TGA-TGG-AGG-AG-3'

   WT Forward: 5'-CAC TTG CTC TCC CAA AGT CGC TC-3'
   WT Reverse: 5'-ATA CTC CGA GGC GGA TCA CAA-3'

2) _Ksp_-Cre- Transgene Forward: 5'-GCA GAT CTG GCT CTC CAA AG-3'

   Transgene Reverse: 5'-AGG CAA ATT TTG GTG TAC GG-3'

3) - _Pax8-CreERT_ -Transgene Forward: 5'-TGC CAC GAC CAA GTG ACA GCA ATG-3'

   Transgene Reverse: 5'-ACC AGA GAC GGA AAT CCA TCG CTC-3'

4) _PRCC-TFE3^{LSL}_ - Transgene Forward: 5'-TTC CCC TCG TGA TCT GCA AC-3'

   Transgene Reverse: 5'-CTG GAA AGA CCG CGA GAG AGT-3'
   WT Forward: 5'-TTC CCC TCG TGA TCT GCA AC-3'
   WT Reverse: 5'-TCA TGG AAA TCT CCG AGG CG-3'

e) _BUN and Creatinine measurements:_ were performed using mouse serum by IDEXX BioAnalytics. **f)** _BrdU labeling_: To measure in vivo proliferation, mice were injected i.p. with a single dose of BrdU (550891, BD Pharmingen), at a dose of 100 mg/kg body weight, 3 hours prior to sacrificing. BrdU incorporation was detected by immunohistochemistry of paraffin-embedded sections using an anti-BrdU monoclonal antibody (5292 CST; 1:200).

**Histology and immunostaining.** Mouse kidneys were fixed in 10% neutral buffered formalin (Sigma-Aldrich), embedded in paraffin, sectioned at 4 μm and used for H&E staining and immunohistochemistry. TFE3, GPNMB, Melan A, PMEL, Phospho-S6 Ribosomal Protein (Ser235/236), Phospho-4E BP1 (Thr37/46), TFEB, PAX8, PAX2, Pan-Keratin, CK8, Vimentin α-SMA, Synaptophysin, BrDU, Ki67, and Phospho-Histone 3 IHC on murine tissues was performed on the Ventana Discovery ULTRA (version v12.31) (Ventana/ Roche) using hand-applied antibodies at the following concentrations: **TFE3** (Invitrogen PA5-54909; 1:5000), **GPNMB** (#90205 CST; 1:100)**, Melan A** (#ab210546 Abcam; 1:500), **PMEL** (#ab137078 Abcam; 1:100), **Phospho-S6 Ribosomal Protein (Ser235/236)** (#4858 CST; 1:200), **Phospho-4E BP1 (Thr37/46)** (#2855 CST; 1:800), **TFEB** (#A303-673A, Bethyl; 1:1000), **PAX8** (#ab191870 Abcam; 1:100)**, PAX2** (#ab79389 Abcam; 1:500)**, Pan-Keratin** (#83957 CST; 1:25), **CK8** (#ab53280 Abcam; 1:100), **Vimentin** (#5741 CST; 1:200), **α-SMA** (#19245 CST; 1:200), **Synaptophysin** (#36406 CST; 1:50), **BrdU** (#5292 CST; 1:200), **Ki67** (#12202 CST; 1:100), **p-Histone H3** (#06-570 Sigma; 1:500).

**Quantification of nuclear Pax8 and Pax2 in murine renal tumors.** Immuno-stained slides were digitally scanned (Nanozoomer, Hamamatsu). Using _STK_ and _PTK_ kidney dual stained slides (GPNMB with Teal and PAX8 with DAB), we trained the HALO Tissue Classifier (IndicaLabs) to attribute classes (GPNMB-positive tubules or GPNMB-negative tubules) based on the stains and textures of the tissue and generated annotations accordingly. A CytoNuclear v2.0.9 algorithm (HALO®, Indica Labs) was run in each annotation to detect PAX8-negative and -positive tubular cells and further grade the positive cells into three

intensities. The resulted H-score was used to compare the PAX8 intensity between the two classes. To validate the classifier, the algorithm was run in manually annotated areas and its results were correlated to the auto-detected annotations (r:0.97, p < 0.000). The analysis algorithm was visually validated. PAX2 staining was visually assessed and showed the same pattern of positivity and intensity as PAX8.

**LTL/DBA Immunofluorescence.** LTL/DBA staining was performed on paraffin-embedded kidney sections that were deparaffinized and rehydrated according to standard histological protocols. Sections were immersed in Target Retrieval Solution (Dako, S169984-2), heated in a steamer for 20 min and blocked using a blocking buffer containing 5% fetal bovine serum (FBS), 0.5% bovine serum albumin (BSA), and 0.1% Triton X-100 for 30 minutes. Rhodamine-labeled Dolichos biflorus agglutinin (DBA) and fluorescein-labeled Lotus tetragonolobus lectin (LTL) (Vector Laboratories, RL-1032 and #FL-1321, respectively) at 1:200 dilution, were incubated overnight at 4 °C. Subsequently, the nuclei were counterstained with DAPI, followed by three washes with PBST (PBS with Tween-20). Finally, the sections were mounted using Fluoromount-G™ Mounting Medium (Invitrogen, #00-4958-02). Images were captured using a Nikon W-1 spinning disk confocal microscope at the UMB-SOM Confocal Microscopy Core in Baltimore, Maryland.

**H&E of a human TFE3-rearranged PEComa.** This is a diagnostic H&E-stained slide from a de-identified patient provided as an example of the histology of human PEComas. It is excess diagnostic material obtained under a waiver of consent and is fully de-identified. Thus, gender and age are not provided. IRB number: IRB00223370.

**In Vivo drug studies and tumor burden estimation.** Treatment of _SFPQ-TFE3^{LSL}; Pax8-CreERT_ with vehicle or Torin1 was initiated at 7-8 weeks following induction of _SFPQ-TFE3_ expression with tamoxifen, for a duration of 4 weeks, across 4 separate cohorts. For each cohort, following tamoxifen treatment, _STP_ mice were allocated into 2 groups-vehicle or Torin1. Torin1 was dissolved in 100% N-methyl-2-pyrrolidone (443778, Sigma-Aldrich) at a stock concentration of 6.25 mg/ml, subsequently diluted 1:4 with sterile 50% PEG400 (06855- Sigma-Aldrich) to a final concentration of 1.25 mg/ml and delivered intraperitoneally at 5/10 mg/kg, once daily, for 5 days a week. Mouse weights were recorded weekly to assess drug toxicity. Kidney tumors were harvested for histological evaluation and assessment of tumor content. FFPE sections of the kidneys were stained for GPNMB and slides scanned as described above. Each kidney section was annotated as a region of interest, and tumor area was measured via an automated tissue classifier using HALO® trained to detect the GPNMB stain. The percentage of area positive for GPNMB was calculated and compared between vehicle and Torin1-treated groups for each cohort. For all tumor studies, tumor size did not exceed the maximal tumor size/burden permitted by the Johns Hopkins Animal Care and Use Committee (ACUC) (maximum tumor size for a single spontaneous or implanted tumor tumor that is visible without imaging ~2 cm in any dimension in adult mice).

**Cell and tissue lysates and immunoblotting.** Renal tumors: were homogenized and lysed using the gentleMACS M Tubes/ Octo Dissociator in ice-cold RIPA lysis buffer. Cells were lysed in RIPA buffer or cell lysis buffer (for p-TFEB expression) (9803, Cell Signaling). All lysis buffers were supplemented with 10 μl Halt Protease and Phosphatase Inhibitor Cocktail (78440, Thermo Fisher Scientific). Lysates were centrifuged at $21,380 \times g$ for 10 min at 4 °C and supernatants collected. Protein concentrations were quantified using the BCA Protein Assay Kit (23225, Pierce) using the xMark Microplate Spectrophotometer and Microplate Manager Software (version 6.3) (Biorad), and protein was resolved on 4–12% Bis-Tris

SDS-PAGE gel (Thermo Fisher Scientific). Protein was transferred to nitrocellulose membranes (Amersham Bioscience), blocked for 1 h at room temperature in 5% nonfat milk in 1X TBS-T and then incubated overnight with a primary antibody diluted in 5% BSA or milk in 1X TBS-T. The secondary antibodies used were anti-rabbit or anti-mouse immunoglobulin as appropriate (Cell Signaling) and diluted at 1:1000 in 5% nonfat milk in 1X TBS-T. Blots were developed using a chemiluminescent development solution (Super Signal West Femto, Pierce) and bands were imaged on a chemiluminescent imaging system (ChemiDoc Touch imaging System using the ImageLab Touch Software (version 2.3.0.07) (Bio-Rad) or MicroChemi Chemiluminescent imager using the GelCapture Software (version 2.2.2.0) (FroggaBio Inc.). Digital images were quantified using Image J (version 1.52p) and all bands were normalized to their respective β-actin or GAPDH expression levels as loading controls.

Nuclear lysates were prepared using the PARIS kit (AM1921, Thermo Fisher Scientific) according to manufacturer's instructions. Digital images were quantified using Image J and all bands were normalized to their respective Lamin, Histone H3 or Fibrillarin levels as loading controls. Statistical analysis was performed using Student's unpaired t-test or one-way ANOVA.

Lysosomal fractionation studies were carried out as previously described[19].

**RNA isolation and quantitative real-time RT-PCR.** Total cellular RNA was extracted using RNeasy Mini kit (74104, Qiagen) according to manufacturer's instructions. RNA was converted to cDNA using SuperScript III First-Strand Synthesis System (18080051, Thermo Fisher Scientific) according to manufacturer's instructions. mRNA levels were quantified using the StepOnePlus Real-time PCR system and software (version 2.3) (Applied Biosystems) with the following human primers and probes: PAX2 (Hs01057416_m1), PAX8 (Hs00247586_m1), HNF1B (Hs01001602_m1), LHX1 (Hs00232144_m1), GATA3 (Hs00231122_m1), PMEL (Hs00173854_m1), GPNMB (Hs01095669_m1) and GAPDH (Hs02786624_g1) and mouse primers and probes: RRAGC (Mm00600306_m1), RRAGD (Mm00546741_m1), GAPDH (Mm99999915_g1), ATP6V1A (Hs01097169_m1), ATP6V1B2 (Hs00156037_m1), ATP6V0C (Hs00798308_sH), ATP6V1C1 (Hs00940702_m1), ATP6V0D1 (Hs00371517_m1), ATP6V1D (Hs00211133_m1), ATP6V1E1 (Hs00762211_s1), ATP6V1G1 (Hs00895280_g1), ATP6V1H (Hs00977530_m1), ATP6V0D2 (Hs01084784_m1). *TFE3* fusion transcripts were detected using the following primer pairs, as previously described[28] for: a) *SFPQ-TFE3* -SFPQ exon 7 primer 5′-CGTCAACGT GAGATGGAAGA-3′ (forward primer) -exon 6 *TFE3* primer 5′-GCAG GAGTTGCTGACAGTGA-3′ (reverse primer), and b) *PRCC-TFE3* -PRCC exon 1 primer, 5′-AGGAAAGAGCCCGTGAAGAT-3′ (forward primer) and TFE3 exon 6 primer, 5′-GTTCTCCAGATGGGTCTGC-3′ (reverse primer). Threshold cycle (Ct) was obtained from the PCR reaction curves and mRNA levels were quantitated using the comparative Ct method with actin or Gapdh mRNA serving as the reference. Statistical analysis was performed using Student's unpaired t-test.

**ChIP-seq and ChIP-PCR.** Doxycycline-inducible, HK2/*SFPQ-TFE3* cells expressing HA-tagged SFPQ-TFE3 were cultured with Doxycycline for 24 h, followed by crosslinking with 1% formaldehyde at room temperature for 5 min, and incubation with 125 mM glycine. Chromatin immunoprecipitation and ChIP-Sequencing were performed using SimpleChIP Plus Enzymatic Chromatin IP Kit (Magnetic Beads; 9005, Cell Signaling, Danvers, MA, USA) according to the manufacturer's protocol, and as previously described[102].

**ChIP-seq analysis**
ChIP-Seq Analysis was carried out as previously described[103]. Peak detection was performed using the MACS2 algorithm[104] in the Strand NGS software (Strand Life Sciences). HA-SFPQ-TFE3 binding was identified by significant enrichment of each signal over input DNA peaks with a p-cutoff value of $10^{-5}$. ChIP-seq data visualization was carried out using deepTools v3.4.1[105]. Coverage tracks were generated with the bamCoverage tool, normalized to Reads Per Genomic Content (RPGC). The resulting tracks were visualized using Gviz[106]. For consistency, ENCODE4 data were processed and visualized using the same workflow, starting from publicly available alignment files[107].

In ChIP qPCR, DNA purification was performed in the same manner as in ChIP seq. The primer sequences utilized in the qPCR assays are listed as follows:
1) **GPNMB:**
   5′-CTCCTTGAATCTAGTAAATGTGG-3′
   3′-CAGGGCCTCGGGGAGGATC-5′
2) **PAX8 22831-1**
   5′-GTGATCAGGATAGCTGCCGAC-3′
   3′-CTGCTCGCCTTTGGTGTGGC-5′
3) **PAX8 22831-2**
   5′-CAACCAGATGAGGAGTGTCTTC-3′
   3′-GGGACCAGCAGCCCAGACG-5′
4) **PAX8 22834**
   5′-CGTCAATCCTTTTATCTCCCTC-3′
   3′-GATGTGTCCTGGGCACTTGTC-5′
5) **PAX8 22835**
   5′-GGCCCACTCCCTGGGTTC-3′
   3′-GACAGCCAACTAGGTGGCC-5′
6) **PAX8 14370**
   5′-CAGTCCATGGTTCGGCTTGC-3′
   5′-AGTTGAGCTGGGGACTGCAG-3′

**Immunocytochemistry.** Immunocytochemistry was carried out as previously described[108] and immunofluorescence was visualized using an Olympus BX41 epifluorescence microscope using DP Controller software (version 3.2.1.276) (Olympus, Center Valley, PA).

**Cathepsin B activity assays.** Cathepsin B activity assays were carried out as previously described[108] and analyzed using the GloMax Multi Detection System with Instinct Software (version 3.1.2) (Promega).

**Autophagic flux assays.** To measure autophagic flux, cells were incubated with CYTO-ID® Green Detection Reagent (Enzo) for 30 min and processed according to the manufacturer's instructions for fluorescence plate reader analysis, using the GloMax Multi Detection System with Instinct Software (version 3.1.2) (Promega). Fluorescent intensities were normalized to concurrent supravital staining performed using Hoechst 33342.

**RNA sequencing and data analysis.** RNA sequencing of murine transgenic kidneys was performed at Novogene and carried out as previously described[19]. Raw RNAseq counts were Fragments Per Kilobase of transcript per Million mapped reads (FPKM)-normalized for data visualization in R (v4.3.2). Raw counts were also imputed in DESeq2 in R to determine differentially expressed genes. Log2 fold-changes, p-values, and adjusted *p*-values (false discovery rate method, FDR) were obtained for all genes and comparisons.

**Gene set enrichment analysis (GSEA).** Raw counts were used as input for Gene Set Enrichment Analysis (GSEA, http://www.broad.mit.edu/gsea/). We employed the curated Hallmarks pathways (https://www.gsea-msigdb.org/gsea/msigdb/human/genesets.jsp?collection=H) to identify differential regulation of pathways in our comparison groups. In addition, curated sets from published studies were used to identify significantly enriched pathways in our study groups from data deposited in: a) dbGap under accession code phs001357.v1.p1[36]., b) GEO GSE252047[26] or c) GEO GSE130072[4]. Negative NES indicated

negatively enriched pathways in our comparison group vs. control. Q-value cutoffs were set to 0.1. Pan-Cancer-normalized RNAseq data from TCGA was downloaded from TCGA Pan-Cancer publication portal (https://gdc.cancer.gov/about-data/publications/panimmune). KIRP and KIRC RNAseq data were previously normalized with the methods Fragments Per Kilobase of transcript per Million mapped reads (FPKM) and FPKM Upper Quartile (FPKM-UQ) by the TCGA research team. Gene expression data was compared using Wilcoxon rank sum tests adjusted with multiple comparisons, where applicable. GSEA analyses were conducted with raw RNAseq counts using custom pathways and standard enrichment molecular signatures database. All TCGA graphs presented in the manuscript were generated by us using TCGA data. All analyses were performed in R v4.3.1.

**Statistics and reproducibility.** For RNA (qRT-PCR and ChIP-PCR) quantification, protein quantification and fluorometric quantification of Magic Red and CYTO-ID green fluorescence, statistical significance was determined using the unpaired, two-tailed Student's t-test when comparing two experimental groups, or with one-way ANOVA with Dunnett's or Bonferroni's correction when comparing 3 or more experimental groups. Kaplan-Meier survival analyses were performed by Log-rank (Mantel-Cox) test. Statistical analyses of kidney to body weight ratios, BUN levels, Serum creatinine levels, BrdU incorporation and positivity, Ki67 positivity and Phosphorylated-Histone H3 (pH3) positivity were performed by two-tailed Mann-Whitney test. Digital quantification of median nuclear Pax8 and Pax2 H-scores in murine renal tumors and median GPNMB positivity in tumor burden estimation studies, was analyzed by two-tailed Mann-Whitney test. Gene expression data (FPKM plots) were analyzed by Wilcoxon rank sum test adjusted with multiple comparisons using the false discovery rate (FDR) method. Mean values were performed in GraphPad Prism (version 8.2.1). $p$-values of <0.05 were considered statistically significant.

All experiments in this study were replicated in three or more independent biological replicates with similar results. Additionally, all experiments were performed using multiple litters or cohorts of mice and/or cellular replicates and using multiple orthogonal techniques to ensure rigor. For example: a) SFPQ-TFE3 nuclear localization was confirmed by immunofluorescence and immunoblotting of nuclear-cytoplasmic fractions, b) increased SFPQ-TFE3 transcriptional activity was confirmed by qRT-PCR, immunoblotting and IHC of lysosomal proteins (RAG GTPases, V-ATPases and melanotic markers) c) PAX2/PAX8 nuclear localization was confirmed by immunofluorescence (cells), IHC (mouse renal tumors) and immunoblotting of nuclear-cytoplasmic fractions, d) mTOR activation was assessed in multiple inducible strains and patient derived *TFE3*-fusion cells, as well as three strains of fusion-TFE3 transgenic mice, e) ChIP-Seq peaks of significance were validated by ChIP-PCR, f) In vivo drug studies with Torin1 were performed in multiple cohorts of both, male and female STP transgenic mice.

### Reporting summary
Further information on research design is available in the Nature Portfolio Reporting Summary linked to this article.

## Data availability
All data generated and analyzed during the current study are included in this published article and its supplementary information files, or are deposited in GEO. A reporting summary for this article is available as a Supplementary Information file. The RNA-seq data from this study are deposited into NCBI's Gene Expression Omnibus (GEO) database with the accession code GSE284169 (https://www.ncbi.nlm.nih.gov/geo/query/acc.cgi?acc=GSE284169). The ChIP-Seq data from this study are deposited into NCBI's Gene Expression Omnibus (GEO) database with the accession code GSE297289. Source data are provided with this paper.

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

## Acknowledgements
This research was supported in part by the CDMRP TSCRP grant W81XWH-22-1-0264 (TLL), CDMRP KCRP grant W81XWH-20-1-0843 (TLL), CDMRP KCRP grant W81XWH-22-1-0377 (KA), Kidney Cancer Association KCA-2024-TRANS-001 (KA) and the NCI Cancer Center Support Grant 5P30CA006973-52. This work was supported at Johns Hopkins in part by Dahan Translocation Carcinoma Fund and Joey's Wings Foundation. This work relied on the expertise provided by the Polycystic Kidney Disease Research Resource Consortium, U54 DK126114. This research was supported by the Intramural Research Program of the NIH, National Cancer Institute, Center for Cancer Research (WML). This project has been funded in whole or in part with Federal funds from the National Cancer Institute, National Institutes of Health, under Contract No. HHSN261201500003I. (LSS). The content of this publication does not necessarily reflect the views or policies of the Department of Health and Human Services, nor does mention of trade names, commercial products, or organizations imply endorsement by the U.S. Government.

## Author contributions
T.L.L. and K.A. conceived the study. K.A., A.A., and T.L.L. drafted the manuscript. K.A., A.A., J.W., S.N.A., T.V., E.I., A.S., K.F., H.B.L., M.K., Y.S., M.B., Y.O., P.O., T.W., A.Z.R., L.S.S., W.M.L., P.A., and T.L.L. completed the data collection and analysis. All authors critically reviewed the manuscript and agreed to submit it for publication.

## Competing interests
T.L.L. has received research support from Roche/Ventana, DeepBio, Myriad Genetics, Artera, AIRA Matrix and Exact Biosciences for other studies and consults for AstraZeneca. All other authors report no competing interests.
