## [Transparent Peer Review file · Nature Communications]

SFPQ-TFE3 reciprocally regulates mTORC1 and induces lineage plasticity in a mouse model of renal tumorigenesis

Corresponding Author: Dr Kaushal Asrani

Version 0:

Reviewer comments:

Reviewer #1

(Remarks to the Author)

The authors should be congratulated on an excellent and interesting study. Singh et al present an impressive and high quality study that describes a new mouse model of PEComa/tRCC based on the development and detailed characterization of a transgenic model or the fusion protein SFPQ-TFE3. The study also analyses PRCC-TFE3 mice, other TFE3 fusion proteins in cell culture, human cancer cell lines, as well as human kidney cancer databases. The Cre-inducible SFPQ-TFE3 mice were characterised using two different Cre driver lines to induce renal epithelial cell-specific gene deletion during embryogenesis and during adulthood. Both of these mice developed strong tumour phenotypes which were very convincingly characterised at the level of histology, immunostainings, western blotting and RNA-sequencing. These mouse models highlight the similarities in molecular features between PEComas and tRCC in many ways, including strong loss of multiple markers of epithelial cells, including epithelial cell specifying transcription factors (PAX8, PAX2 and their downstream target transcription factors) and the gain of many molecular markers of tRCC and PEComas. The authors convincingly demonstrate the bidirectional connections between TFE3 fusion protein expression and mTOR pathway activation in vivo and by cell culture experiments using multiple complementary cellular systems involving inducible expression and gene knockdowns/CRISPR-mutations. These studies implicate RRAGC and RRAGD as mediators of increased mTOR activation induced by SFPQ-TFE3 fusion and identify a feed forward regulatory loop between mTOR and TFE3 fusion proteins that lead to the loss of epithelial cell identity and cellular transformation. These studies collectively shed light on the similarities and differences between different TFE3 fusion proteins and their roles in tRCC, as well as in PEComas. An important finding is that these tumours may have the same cell of origin and the different pathologically recognized diseases may in fact be a continuum based on degrees of loss of cellular identity. The results will drive future experiments aiming to more precisely define the molecular connections between MiT/TFE-based oncogenes and their interactions with the mTOR pathway and cellular lineage maintenance.

The manuscript is very well written, clearly explained and the figures are of very high quality. The manuscript also contains a very thoughtful discussion that puts the study into the bigger picture of renal (and other organ) tumorigenesis and highlights the spectrum of phenotypes that can result from different fusion proteins.

I have only one point related to the experiments. Figure 5H is missing an appropriate set of wild type controls. Since primary renal epithelial cells de-differentiate over time in culture, in order to be able to make the conclusion that altered protein levels (eg of PAX8, PAX2, WT-1 etc) are related to the expression of the SFPQ-TFE3 transgene it would be essential to conduct a time course experiment using the same genotypes as in Fig S1B. It is not clear at what timepoint Ad-null was harvested.

Reviewer #2

(Remarks to the Author)

The manuscript entitled "SFPQ-TFE3 gene fusion reciprocally regulates mTORC1 activity and induces lineage plasticity in a novel mouse model of renal tumorigenesis" describes a novel mouse model to investigate SFPQ-TFE3-driven renal tumorigenesis. They also show that the induction of SFPQ-TFE3 leads to downregulating nephric lineage markers (PAX2, PAX8) and promotes a PEComa-like phenotype. Mechanistically, the authors claim that SFPQ-TFE3 expression activates the mTORC1 pathway through increased RRAGC/D transcription. Also, the mTOR inhibition can rescue PAX2/8 expression, reduce TFE3 fusion protein levels, and reverse the lineage plasticity. Thus, these findings highlight a reciprocal

relationship between SFPQ-TFE3 and mTORC1 signaling, providing insights into the pathogenesis and potential therapeutic targets for TFE3-driven tumors.

The data presented are technically sound and obtained using appropriate methodologies. Thus, these results might contribute to advancing our understanding of the signaling underlying translocation-driven renal tumors, opening future avenues for targeted therapies. However, the mechanistic details of how mTORC1 stabilizes TFE3 fusion proteins remain unclear. Also, more functional experiments are needed to link RRAGC/D to the activation of mTORC1. Thus, I recommend the revision of the following aspects before considering this manuscript for publication.

Major concerns

1. Further investigate the mechanism by which mTORC1 stabilizes SFPQ-TFE3 fusion proteins. Some experimental suggestions are: Protein stability studies to determine the half-life of SFPQ-TFE3 in the presence and absence of mTOR inhibitors; Ubiquitination analysis to investigate whether mTOR inhibition promotes ubiquitination and proteasomal degradation of SFPQ-TFE3; Autophagy assays to examine if mTOR inhibition accelerates autophagic degradation of SFPQ-TFE3 by measuring LC3-II conversion and p62 degradation.

2. Clarify the relative contributions of RRAGC and RRAGD to mTORC1 activation. For instance, overexpress wild-type or mutant RRAGC/D to see whether the activation of mTORC1 and SFPQ-TFE3 stability can be restored; Perform immunofluorescence to determine if SFPQ-TFE3 expression affects the localization of mTORC1 to lysosomes in an RRAGC/D-dependent manner.

3. In Vivo Therapeutic Validation with mTOR Inhibitors. Treat SFPQ-TFE3 transgenic mice (STP model) with mTOR inhibitors (e.g., Torin1 or everolimus) and assess: Tumor growth and histopathology; Expression of PAX2/PAX8 and melanocytic markers; Levels of SFPQ-TFE3 fusion protein; Perform a longitudinal study to determine if mTOR inhibition improves survival in SFPQ-TFE3 transgenic mice.

4. Lineage Plasticity Mechanism. The manuscript shows that SFPQ-TFE3 drives lineage plasticity, but the mechanism of PAX2/PAX8 downregulation is not fully explained. Consider performing ChIP-Seq for SFPQ-TFE3 to identify direct targets and determine if it binds to the promoters of PAX2/PAX8; Investigate changes in histone modifications (e.g., H3K27ac, H3K27me3) at the PAX2/PAX8 loci after SFPQ-TFE3 induction; Perform ATAC-Seq to assess changes in chromatin accessibility around PAX2/PAX8 regulatory regions.

Minor concerns

1. Provide a summary of the clinical relevance of PEComas and RCC in the introduction.
2. Include a better and more self-explaining schematic diagram summarizing the key findings and mechanisms.

Reviewer #3

(Remarks to the Author)

Asrani et al. report the generation and characterization of a new kidney-specific TFE3 fusion transgene mouse model, which recapitulates SFPQ-TFE3-linked tumor phenotypes seen in human tumors. The work is comprehensive and appears carefully conducted. The results are clear and of clinical interest. The writing and experimentation is thoughtful. Overall, the work is very interesting. I have only a couple of suggestions / comments.

The variation in renal lineage factor expression between the different mouse and human TFE3 fusion models indicates possible genotype-phenotype correlations. Are there any differences in the TFE3-fusions between tRCCs and PEComas? It would be helpful to summarize such data somehow, maybe even in a table. Could the recent paper by Achom et al. be of help here (PMID: 39168126)?

How do the authors think mTORC1 is activated downstream of TFE3 fusions? mTOR hyperactivation has been linked to other forms of RCC as well via auto/paracrine signaling or via mutations (PMID: 28473526, 30858363, 38861592). This is particularly relevant as the authors suggest that mTOR activation in the tRCC context can lead to PAX8 inhibition in an mTOR-dependent manner. This raises the question, why doesn't mTOR downregulate PAX8 and cause lineage switching in all RCCs? Here the different TFE3 fusions could be helpful. It is understood that answering these questions could be experimentally challenging and probably outside the scope of the present manuscript, but it would be good to discuss this point to avoid the impression of a direct link from mTOR activation to loss of renal lineage markers.

Minor:

The manuscript describes strikingly many striking results.

Line 541, while PAX8 promotes oncogenic signaling, it is perhaps not best characterized as an oncogene.

Figure 2E, are all the panels of the same magnification? Also, Figure 2H has two scale bars but many panels.

Instead of presenting tables in a PDF (>2,000 pages!), I would consider alternative file formats.

Reviewer #4

(Remarks to the Author)

Summary

The authors present a new transgenic mouse model where the induction of an SFPQ::TFE3 fusion leads to the development

of renal PEComas, both morphologically and immunophenotypically. They use two types of mice: (1) one where the fusion is introduced during embryogenesis, causing death within 20 days but leading to PEComa-like tumors around day 15 that express melanocytic markers, and (2) a conditional model (activated by tamoxifen) where PEComa-like tumors develop after 8-12 weeks. The study shows that induction of a MiT/TFE fusion (specifically SFPQ::TFE3) increases MTORC1 activity and is associated with a loss of PAX8, PAX2, and keratin expression. The authors also validate these findings using human cell lines. Finally, they demonstrate that inhibiting mTOR signaling (genetically or pharmacologically) reduces the expression of the SFPQ::TFE3 fusion protein and restores nephric lineage marker expression and transcriptional activity in vitro.

Novelty

- First transgenic mouse model for SFPQ::TFE3 fusion
- Downregulation of renal lineage markers is most dramatic in this mouse as compared to other models such as PRCC::TFE3
- First evidence that PEComas may derive from an epithelial cell (the same origin as tRCC)

Strengths

- Methodology is good, a combination of mouse and human models; experiments are performed with appropriate number of animals to achieve power
- Detailed methods are provided
- Work/results support the hypothesis
- Relevant references and prior literature are addressed, findings are coherent with published work (i.e. SFPQ::TFE3 is the commonest fusion in renal PEComa, includes all relevant papers of 2024)

Comments/questions

-
- Does the morphology change once mTOR is inhibited?
- Do we need to distinguish between both neoplasms? While earlier studies (Wang et al, Human Path 2017) seem to try to describe the differences, this may not matter? Clinical relevance might still be the treatment with mTOR inhibitors? This seems to be something that should be addressed in the discussion.

Reviewer #5

(Remarks to the Author)

Version 1:

Reviewer comments:

Reviewer #1

(Remarks to the Author)

Thank you for carrying out the new experiment that I suggested and including this as a new panel in Suppl Fig 8F. This experiment completely answers the question that I had. I congratulate the authors once again on this great study!

Reviewer #2

(Remarks to the Author)

I wanted to express my appreciation for the effort of the authors in responding to all the criticisms and suggestions of my review. The authors have responded to all these questions which I believe have substantially improved this article. I have no further questions and therefore recommend this article for publication.

Reviewer #3

(Remarks to the Author)

The authors have addressed all my previous points, I have no further comments.

Reviewer #4

(Remarks to the Author)

All our comments have been adequately answered and addressed.

Reviewer #5

(Remarks to the Author)

REVIEWER COMMENTS

Reviewer #1 (Remarks to the Author):

Reviewer Comment: The authors should be congratulated on an excellent and interesting study. Singh et al present an impressive and high quality study that describes a new mouse model of PEComa/tRCC based on the development and detailed characterization of a transgenic model or the fusion protein SFPQ-TFE3. The study also analyses PRCC-TFE3 mice, other TFE3 fusion proteins in cell culture, human cancer cell lines, as well as human kidney cancer databases. The Cre-inducible SFPQ-TFE3 mice were characterised using two different Cre driver lines to induce renal epithelial cell-specific gene deletion during embryogenesis and during adulthood. Both of these mice developed strong tumour phenotypes which were very convincingly characterised at the level of histology, immunostainings, western blotting and RNA-sequencing. These mouse models highlight the similarities in molecular features between PEComas and tRCC in many ways, including strong loss of multiple markers of epithelial cells, including epithelial cell specifying transcription factors (PAX8, PAX2 and their downstream target transcription factors) and the gain of many molecular markers of tRCC and PEComas. The authors convincingly demonstrate the bidirectional connections between TFE3 fusion protein expression and mTOR pathway activation in vivo and by cell culture experiments using multiple complementary cellular systems involving inducible expression and gene knockdowns/CRISPR-mutations. These studies implicate RRAGC and RRAGD as mediators of increased mTOR activation induced by SFPQ-TFE3 fusion and identify a feed forward regulatory loop between mTOR and TFE3 fusion proteins that lead to the loss of epithelial cell identity and cellular transformation. These studies collectively shed light on the similarities and differences between different TFE3 fusion proteins and their roles in tRCC, as well as in PEComas. An important finding is that these tumours may have the same cell of origin and the different pathologically recognized diseases may in fact be a continuum based on degrees of loss of cellular identity. The results will drive future experiments aiming to more precisely define the molecular connections between MiT/TFE-based oncogenes and their interactions with the mTOR pathway and cellular lineage maintenance.

The manuscript is very well written, clearly explained and the figures are of very high quality. The manuscript also contains a very thoughtful discussion that puts the study into the bigger picture of renal (and other organ) tumourigenesis and highlights the spectrum of phenotypes that can result from different fusion proteins.

Author Response: We thank the reviewer for their positive feedback.

Reviewer Comment: I have only one point related to the experiments. Figure 5H is missing an appropriate set of wild type controls. Since primary renal epithelial cells de-differentiate over time in culture, in order to be able to make the conclusion that altered protein levels (eg of PAX8, PAX2, WT-1 etc) are related to the expression of the SFPQ-TFE3 transgene it would be essential

to conduct a time course experiment using the same genotypes as in Fig S1B. It is not clear at what timepoint Ad-null was harvested.

Author Response: We thank the reviewer for their positive feedback. Regarding Figure 5H, the Ad-null samples in lane 1 were harvested on day 5, and this has now been added to the figure legend. We have now provided an additional time course experiment of *SFPQ-TFE3^{LSL}* primary renal cells treated with Ad-null and Ad-cre adenoviruses, harvested on days 2, 4, 6 and immunoblotted for TFE3 and renal lineage markers. These new results are in **Fig. S8F** of the revised manuscript.

Reviewer #2 (Remarks to the Author):

Reviewer Comment: The manuscript entitled “SFPQ-TFE3 gene fusion reciprocally regulates mTORC1 activity and induces lineage plasticity in a novel mouse model of renal tumorigenesis” describes a novel mouse model to investigate SFPQ-TFE3-driven renal tumorigenesis. They also show that the induction of SFPQ-TFE3 leads to downregulating nephric lineage markers (PAX2, PAX8) and promotes a PEComa-like phenotype. Mechanistically, the authors claim that SFPQ-TFE3 expression activates the mTORC1 pathway through increased RRAGC/D transcription. Also, the mTOR inhibition can rescue PAX2/8 expression, reduce TFE3 fusion protein levels, and reverse the lineage plasticity. Thus, these findings highlight a reciprocal relationship between SFPQ-TFE3 and mTORC1 signaling, providing insights into the pathogenesis and potential therapeutic targets for TFE3-driven tumors.

The data presented are technically sound and obtained using appropriate methodologies. Thus, these results might contribute to advancing our understanding of the signaling underlying translocation-driven renal tumors, opening future avenues for targeted therapies.

Author Response: Thank you for this positive appraisal of our study.

Reviewer Comment: However, the mechanistic details of how mTORC1 stabilizes TFE3 fusion proteins remain unclear. Also, more functional experiments are needed to link RRAGC/D to the activation of mTORC1. Thus, I recommend the revision of the following aspects before considering this manuscript for publication.

Author Response: Please see detailed point-by-point responses below.

Major concerns

Reviewer Comment: 1. Further investigate the mechanism by which mTORC1 stabilizes SFPQ-TFE3 fusion proteins. Some experimental suggestions are: Protein stability studies to determine the half-life of SFPQ-TFE3 in the presence and absence of mTOR inhibitors; Ubiquitination analysis to investigate whether mTOR inhibition promotes ubiquitination and proteasomal degradation of SFPQ-TFE3; Autophagy assays to examine if mTOR inhibition accelerates autophagic degradation of SFPQ-TFE3 by measuring LC3-II conversion and p62 degradation.

Author Response: We thank the reviewer for their positive and constructive feedback. As suggested, we have now performed additional experiments investigating the mechanisms by which mTORC1 may stabilize SFPQ-TFE3 fusion proteins. Protein stability studies for SFPQ-TFE3 fusion proteins with cycloheximide treatment did not reveal a difference between vehicle- and torin-treated samples (**Fig. S10A, B**). We also employed a fluorescent spectrophotometric approach to quantitatively estimate autophagic flux or intracellular Cathepsin B activity upon mTOR inhibition *in vitro* (**Fig. S10C, D**), and also examined blocking autophagic flux with BafilomycinA1 (**Fig. S10E**). These experiments did not substantiate a role for increased autophagic degradation of SFPQ-TFE3 with mTOR inhibition. Similarly, co-immunoprecipitation experiments demonstrated a slight decrease (rather than increase) in Ub-SFPQ-TFE3 proportional to the decrease in total protein levels with mTOR inhibition (**Fig. S10F**), and proteosomal inhibition with MG132 or Bortezomib did not reverse the torin1-mediated decrease in SFPQ-TFE3 protein levels (**Fig. S10G**). Taken together, these experiments did not support altered protein stability as a mechanism of SFPQ-TFE3 downregulation with mTOR inhibition. Consistent with these negative data, we show that mTORC1 inhibition with Torin1 or *RHEB* siRNA significantly reduces *SFPQ-TFE3* mRNA levels in HK2/*SFPQ-TFE3* cells (**Fig. S10H**), and *PRCC-TFE3* mRNA levels in UOK124 cells (**Fig. S10I**), indicating that mTORC1 activation regulates *SFPQ-TFE3* fusion expression via a transcriptional mechanism. **These new results are discussed on pages 21-22 (lines 495-521) and page 30 (lines 700-704), and in Fig. S10 of the revised manuscript.**

Reviewer Comment: 2. Clarify the relative contributions of RRAGC and RRAGD to mTORC1 activation. For instance, overexpress wild-type or mutant RRAGC/D to see whether the activation of mTORC1 and SFPQ-TFE3 stability can be restored; Perform immunofluorescence to determine if SFPQ-TFE3 expression affects the localization of mTORC1 to lysosomes in an RRAGC/D-dependent manner.

Author Response: As suggested, we have now performed additional experiments examining the effects of inactive mutants of RRAGC and RRAGD on mTOR activation, SFPQ-TFE3 protein levels and PAX8 expression. We have also tested the effects of SFPQ-TFE3 expression with or without RRAG C/D siRNA on lysosomal localization of Raptor - an essential component of mTORC1- by lysosomal-fraction immunoblotting. Similar to RRAGC/D knock-down (**Fig. S5H**), transient expression of inactive RRAGC/D mutants weakly suppressed mTOR activation but did not rescue SFPQ-TFE3 levels or downstream PAX8 expression (**Fig. S6A**). Consistent with this, while lysosomal levels of Raptor were increased with SFPQ-TFE3 induction (**Fig. S6B**), RRAGC/D siRNA only variably decreased this lysosomal enrichment of Raptor (**Fig. S6C, D**). Taken together, our results suggest that RRAGC and/or RRAGD inactivation, via siRNA or transient expression of inactivating mutants, results in an incomplete suppression of mTORC1 signaling in tRCC, and consequently fails to downregulate fusion-*TFE3* expression (**Fig. S5H, S6A**) or rescue PAX8 expression (**Fig. S6A**). **These new results are discussed on page 13 (lines 296-315), page 21 (lines 488-494) and page 29 (lines 670-674) of the revised manuscript.**

We fully agree that the magnitude of the contribution of RRAGC/D to mTORC1 activation upon SFPQ-TFE3 expression is not clear, and we acknowledged this complexity in our original manuscript, stating that "Taken together, our results suggest that RRAGC and/or RRAGD may

contribute to increased mTORC1 activity in tRCC, but are likely not the only mechanism leading to increased activity of this oncogenic signaling pathway.” In the revised manuscript, we now present an additional mechanism that may contribute to mTOR activation. In a **new Figure 4**, we now demonstrate that expression of multiple subunits of the vacuolar H⁺-ATPase (v-ATPase) complex (an MiT/TFE transcriptional target and an important component of the lysosomal machinery that activates mTORC1) are elevated in human tRCC cases in the TCGA (**Fig. 4A**), and HK2/*SFPQ-TFE3* cells by qRT-PCR (**Fig. 4B**) and/or immunoblotting (**Fig. 4C**), in an mTOR-dependent manner. We show similar data for UOK cells (**Fig. 4E**). Genetic and/or pharmacological inhibition of *ATP6V0C* - an evolutionarily conserved subunit of the complex - via siRNA (**Fig. 4F, 7I**) or Bafilomycin A1 (**Fig. 4G, S6E, S9G, S9H**) is sufficient to decrease phosphorylation of multiple mTORC1 substrates, downregulate *SFPQ-TFE3* expression and rescue *PAX8* expression. Taken together, these data substantiate an important role for V-ATPase complex in mediating increased mTORC1 activation downstream of *SFPQ-TFE3* expression, and this may add to the impact of increased *RRAGC/D* expression. **These new results are discussed on page 14 (lines 318-334), page 21 (lines 485-494) and page 29 (lines 679-685), and in Figs. 4, 7I, S6E, S9G-H of the revised manuscript.**

Reviewer Comment: 3. In Vivo Therapeutic Validation with mTOR Inhibitors. Treat *SFPQ-TFE3* transgenic mice (STP model) with mTOR inhibitors (e.g., Torin1 or everolimus) and assess: Tumor growth and histopathology; Expression of *PAX2/PAX8* and melanocytic markers; Levels of *SFPQ-TFE3* fusion protein; Perform a longitudinal study to determine if mTOR inhibition improves survival in *SFPQ-TFE3* transgenic mice.

Author Response: We examined the effects of Torin1 in reducing kidney tumor burden in *STP* mice. Following induction of *SFPQ-TFE3* expression with tamoxifen, we treated male and female cohorts of mice with vehicle or Torin1 for 4 weeks followed by IHC for p-S6 and the melanocytic marker GPNMB. We utilized an automated tissue classifier algorithm using HALO software to do a detailed quantification of microscopic tumor burden by assessing the percent tissue area expressing GPNMB. There was a small but statistically significant decrease in tumor burden in male, but not female *STP* mice treated with Torin1, potentially due to more rapid tumor growth observed in female versus male mice in the *STP* model at this timepoint. **These new results are discussed on pages 22-23 (lines 522-529) and page 30 (lines 692-695), and in Fig. S11 of the revised manuscript.**

Reviewer Comment: 4. Lineage Plasticity Mechanism. The manuscript shows that *SFPQ-TFE3* drives lineage plasticity, but the mechanism of *PAX2/PAX8* downregulation is not fully explained. Consider performing ChIP-Seq for *SFPQ-TFE3* to identify direct targets and determine if it binds to the promoters of *PAX2/PAX8*; Investigate changes in histone modifications (e.g., H3K27ac, H3K27me3) at the *PAX2/PAX8* loci after *SFPQ-TFE3* induction; Perform ATAC-Seq to assess changes in chromatin accessibility around *PAX2/PAX8* regulatory regions.

Author Response: As suggested, we have now performed chromatin immunoprecipitation sequencing (ChIP-seq) in 2 independent clones of doxycycline-inducible, HA-tagged, HK2/*SFPQ-TFE3* cells (PSF6_8_12 and PSF6_9_9). We observed strong enrichment of *SFPQ-*

TFE3 binding in the *PAX8* gene/body region around Exon 7 and upstream of the *PAX8* TSS. ChIP-qPCR using primers specific for the indicated peaks confirmed binding of *SFPQ-TFE3* to these regions of *PAX8* following doxycycline treatment in HK2/SFPQ-TFE3 cells. In nephron organoids, these peaks overlap with active histone modifications like H3K27ac, typically associated with activation of enhancers. Notably, SFPQ, by recruiting of mSin3A and HDACs, can also function as a transcriptional co-repressor via gene deacetylation, thus potentially linking *SFPQ-TFE3* binding to *PAX8* gene repression. These new data are in **Fig. S8H, I** and **Supplementary Tables S7-S12**, and are discussed in results on page 19 (lines 436-450) and discussion on page 28 (lines 653-661) of the revised manuscript.

Minor concerns

Reviewer Comment: 1. Provide a summary of the clinical relevance of PEComas and RCC in the introduction.

Author Response: We have now added additional clinical details for tRCC and PEComas in the Introduction on page 3 (lines 74-76 and lines 82-85). In addition, we have now added a table summarizing the key differences between PEComas and RCCs with respect to TFE3-fusions, morphology, immunohistochemical markers, differential diagnoses, therapeutic options and prognosis as **Supplementary Table S13** and on page 25 (lines 585-590) of the revised manuscript (also see Reviewer #3 response, below).

Reviewer Comment: 2. Include a better and more self-explaining schematic diagram summarizing the key findings and mechanisms.

Author Response: Thank you for this suggestion. We have now provided a simplified schematic diagram summarizing the key findings, with references to figures in the manuscript and citations to previous studies, in new **Figure 8** and discussed on page 23 (lines 529-537) of the revised manuscript.

Reviewer #3 (Remarks to the Author):

Reviewer Comment: Asrani et al. report the generation and characterization of a new kidney-specific TFE3 fusion transgene mouse model, which recapitulates SFPQ-TFE3-linked tumor phenotypes seen in human tumors. The work is comprehensive and appears carefully conducted. The results are clear and of clinical interest. The writing and experimentation is thoughtful.

Overall, the work is very interesting. I have only a couple of suggestions / comments.

Author Response: Thank you for this positive appraisal of our study.

Reviewer Comment: The variation in renal lineage factor expression between the different mouse and human TFE3 fusion models indicates possible genotype-phenotype correlations. Are there any differences in the TFE3-fusions between tRCCs and PEComas? It would be helpful to summarize such data somehow, maybe even in a table. Could the recent paper by Achom et al. be of help here (PMID: 39168126)?

Author Response: *SFPQ* is the most common fusion partner in *TFE3*-rearranged PEComas, both in the kidney (52%) as well as extra-renal (58%) sites¹. *SFPQ* is also the third most frequent one in *TFE3*-rearranged renal cell carcinoma, after *ASPSCR1* and *PRCC*. The key differences between PEComas and RCCs with respect to *TFE3*-fusions, morphology, immunohistochemical markers, differential diagnoses, therapeutic options and prognosis are summarized below (adapted from¹⁻⁶). We have now added the following Table as **Supplementary Table S13** and on page 25 (lines 585-590) of the revised manuscript.

		TFE3-rearranged PEComa	TFE3-rearranged RCC
TFE3 fusion partner		SFPQ> NONO> ASPSCR1	ASPSCR1> PRCC> SFPQ
Morphology:		Pseudorosettes, subnuclear vacuolization, psammomatous calcification Absent	Pseudorosettes subnuclear vacuolization, psammomatous calcification Present
		Nested, pure epitheloid	Papillary architecture (most common) or alveolar, nested; voluminous cytoplasm, discrete cell borders
Marker Expression			
Epithelial	Pan Renal: PAX8	Negative (Absent in 100%)	Positive (Present in 91%)
	Cytokeratins -PanKeratins (AE1/AE1) -CK7	Negative (Absent in 100%)	Patchy/focal (Present in 13-45%)
	Proximal Tubular: CD10	Negative (Absent in 100%)	Positive (Present in 80%)
Muscle	-Desmin -SMA	Absent in 97% Absent in 85%	Absent Absent
Melanotic	-CTSK	Present in 97%	Present in 43%
	-HMB45	Present in 89%	Present in 34%
	-MelanA	Present in 57%	Present in 36%
	-CD68	Present in 100%	Absent in 100%

Differential diagnosis	a) PEComas with biallelic TSC1/2 loss (TFE3 -rearranged PEComas show a tendency to young age, absence of association with TSC syndrome and minimal staining for muscle markers), b) TFE3 -rearranged RCC, c) Xp11 neoplasm with melanocytic differentiation, d) melanotic Xp11 translocation RCC, e) ASPS	a) ccRCC, b) PRCC, c) ccPRCC, d) TFEB -rearranged RCC, e) eAML
Therapeutic options and Response to Rapalogs	Combinations of chemotherapy, VEGF Tyrosine Kinase inhibitors (TKI), and mTOR inhibitors	Combinations of VEGF Tyrosine Kinase inhibitors (TKI), immunotherapy (IO), and mTOR inhibitors
	Variable responses to rapalogs compared to TSC1/2 null-PEComas which show exceptional response. TFE3 -rearranged PEComa/tRCC respond better to mTOR kinase inhibitors than rapalogs.	
Prognosis	Melanotic Xp11 Tumors are more aggressive with poor prognosis. Recurrence/ metastasis in 62.5%	Poor (ASPS/CR1) Intermediate (PRCC) Excellent (SFPQ) Recurrence/ metastasis in 12.5%

Reviewer Comment: How do the authors think mTORC1 is activated downstream of *TFE3* fusions?

Author Response: The mTORC1 pathway is consistently activated in human and murine models of tRCC, with transcriptional induction of *RRAGD* been described as one of the initial mechanisms driving this activation⁷. With new experiments suggested by Reviewer 2 (comment 2) above, our aggregate results suggest that while *RRAGC/D* may contribute to mTORC1 activation in tRCC, *RRAGC/D* over-expression is not necessary for this increased signaling. We now demonstrate that *RRAGC* and/or *RRAGD* inactivation, via siRNA or transient expression of inactivating mutants, is insufficient to fully suppress mTORC1 signaling in tRCC (**Fig. S5H and S6**), and also fails to downregulate fusion-*TFE3* expression (**Fig. S5H, S6A**) or rescue *PAX8* expression (**Fig. S6A**). We did acknowledge this complexity in our original manuscript, stating that "Taken together, our results suggest that *RRAGC* and/or *RRAGD* may contribute to increased mTORC1 activity in tRCC, but are likely not the only mechanism leading to increased activity of this oncogenic signaling pathway."

In the revised manuscript, we now present an additional mechanism that may contribute to mTOR activation in tRCC. In a **new Figure 4**, we demonstrate that expression of multiple subunits of the vacuolar H⁺-ATPase (v-ATPase) complex (an MiT/TFE transcriptional target and an important component of the lysosomal machinery that activates mTORC1) are elevated in human tRCC cases in the TCGA (**Fig. 4A**), and HK2/*SFPQ-TFE3* cells by qRT-PCR (**Fig. 4B**) and/or immunoblotting (**Fig. 4C**), in an mTOR-dependent manner. We show similar data for UOK cells (**Fig. 4E**). Genetic and/or pharmacological inhibition of *ATP6V0C* - an evolutionarily conserved

subunit of the complex - via siRNA (**Fig. 4F, 7I**) or Bafilomycin A1 (**Fig. 4G, S6E, S9G, S9H**) is sufficient to decrease phosphorylation of multiple mTORC1 substrates, downregulate SFPQ-TFE3 expression and rescue PAX8 expression. Taken together, these data substantiate an important role for V-ATPase complex in mediating increased mTORC1 activation downstream of SFPQ-TFE3 expression, and this may add to the impact of increased RRAGC/D expression. **These new results are discussed on page 14 (lines 318-334), page 21 (lines 485-494) and page 29 (lines 679-685), and in Figs. 4, 7I, S6E, S9G-H of the revised manuscript.**

Reviewer Comment: mTOR hyperactivation has been linked to other forms of RCC as well via auto/paracrine signaling or via mutations (PMID: 28473526, 30858363, 38861592). This is particularly relevant as the authors suggest that mTOR activation in the tRCC context can lead to PAX8 inhibition in an mTOR-dependent manner. This raises the question, why doesn't mTOR downregulate PAX8 and cause lineage switching in all RCCs? Here the different TFE3 fusions could be helpful. It is understood that answering these questions could be experimentally challenging and probably outside the scope of the present manuscript, but it would be good to discuss this point to avoid the impression of a direct link from mTOR activation to loss of renal lineage markers.

Author Response: As the reviewer has noted, mTOR is frequently activated in human ccRCC cases, where expression of PAX8 is retained and is, in fact, critical for oncogenesis, implying that mTOR activation alone is insufficient to cause loss of PAX8 and lineage switching. We fully agree with statement, and our data even go further to suggest that among tRCC subtypes, some TFE3 fusion genes (eg, SFPQ-TFE3) more dramatically reduce PAX8 expression than others (eg, PRCC-TFE3) in both *in vitro* (**Fig. 6A**) and *in vivo* (**Fig. 5D, F**) systems. We now expand on this by using ChIP-seq to demonstrate that SFPQ-TFE3 binds directly to regulatory regions of PAX8 (**Fig. S8H, I**) and future work will examine ChIP-seq for PRCC-TFE3 to compare this binding among different fusions as the reviewer suggests. Taken together, our data support a model whereby mTOR inhibition rescues PAX8 expression *indirectly*, via reducing SFPQ-TFE3 expression (**Fig. 7A-D, 7F**), which we now demonstrate happens at the transcription level (**Fig. S10H and I**). We hope we have clarified this with a new model figure (**Fig. 8**) **(and discussed on page 23 (lines 529-537))**, indicating the *indirect* manner in which mTORC1 signaling impacts PAX8 levels via reinforcing SFPQ-TFE3 expression.

Minor:

The manuscript describes strikingly many striking results.

Reviewer Comment: Line 541, while PAX8 promotes oncogenic signaling, it is perhaps not best characterized as an oncogene.

Author Response: This has now been changed to "PAX8 promotes oncogenic signaling in ccRCC", **on line 638 of page 27 of the revised manuscript.**

Reviewer Comment: Figure 2E, are all the panels of the same magnification? Also, Figure 2H has two scale bars but many panels.

Author Response: Fig 2E: This has now been corrected to indicate low magnification (top row) and high magnification (bottom row) images.

Fig 2H: The left and right panels have now been separated with their respective scale bars.

Reviewer Comment: Instead of presenting tables in a PDF (>2,000 pages!), I would consider alternative file formats.

Author Response: We have now uploaded all Supplementary tables in Excel format.

Reviewer #4 (Remarks to the Author):

Summary

Reviewer Comment: The authors present a new transgenic mouse model where the induction of an SFPQ::TFE3 fusion leads to the development of renal PEComas, both morphologically and immunophenotypically. They use two types of mice: (1) one where the fusion is introduced during embryogenesis, causing death within 20 days but leading to PEComa-like tumors around day 15 that express melanocytic markers, and (2) a conditional model (activated by tamoxifen) where PEComa-like tumors develop after 8-12 weeks. The study shows that induction of a MiT/TFE fusion (specifically SFPQ::TFE3) increases MTORC1 activity and is associated with a loss of PAX8, PAX2, and keratin expression. The authors also validate these findings using human cell lines. Finally, they demonstrate that inhibiting mTOR signaling (genetically or pharmacologically) reduces the expression of the SFPQ::TFE3 fusion protein and restores nephric lineage marker expression and transcriptional activity in vitro.

Novelty

- First transgenic mouse model for SFPQ::TFE3 fusion
- Downregulation of renal lineage markers is most dramatic in this mouse as compared to other models such as PRCC::TFE3
- First evidence that PEComas may derive from an epithelial cell (the same origin as tRCC)

Strengths

- Methodology is good, a combination of mouse and human models; experiments are performed with appropriate number of animals to achieve power
- Detailed methods are provided
- Work/results support the hypothesis
- Relevant references and prior literature are addressed, findings are coherent with published work (i.e. SFPQ::TFE3 is the commonest fusion in renal PEComa, includes all relevant papers of

2024)

Author Response: Thank you for this positive appraisal of our study.

Comments/questions

Reviewer Comment: Does the morphology change once mTOR is inhibited?

Author Response: Doxycycline-mediated induction of *SFPQ-TFE3* in HK2 cells resulted in a change in morphology from a cobblestone, epithelial pattern to a more mesenchymal appearance, within 48-72 hrs of doxycycline treatment. mTOR inhibition completely reversed this phenotype with a reversion to epithelial morphology, as shown in the phase contrast images below.

HK-2/SFPQ-TFE3

(-) doxycycline (+) doxycycline (+) doxycycline
RHEB1 siRNA

We also have added data examining the effects of Torin1 in reducing kidney tumor burden in *STP* mice. Following induction of *SFPQ-TFE3* expression with tamoxifen, we treated male and female cohorts of mice with vehicle or Torin1 for 4 weeks followed by IHC for p-S6 and the melanotic marker GPNMB. We utilized an automated tissue classifier algorithm using HALO, to calculate % GPNMB positivity and evaluate tumor burden. There was a small but significant decrease in tumor burden in male, but not female *STP* mice treated with Torin1, however this was very subtle and not accompanied by any change in morphology that we could perceive. **These new results are in Fig. S9C and Fig. S11 (A-C) and on page 20 (lines 473-475) pages 22-23 (lines 522-529) and page 30 (lines 692-695) of the revised manuscript.**

Reviewer Comment: Do we need to distinguish between both neoplasms? While earlier studies (Wang et al, Human Path 2017) seem to try to describe the differences, this may not matter? Clinical relevance might still be the treatment with mTOR inhibitors? This seems to be something that should be addressed in the discussion.

Author Response: Our data suggest that tRCC and TFE3-fusion-driven PEComas may exist on a continuum, with PEComas representing cases with the most extreme plasticity. We agree that distinctions between the two entities may be somewhat artificial and perhaps not impactful clinically, though this requires additional study. Because PEComas are relatively enriched for SFPQ-TFE3 fusions compared to tRCC, distinctions between the two neoplasms may actually be driven by the underlying driver fusion partners, rather than the histology. With respect to differences in *SFPQ-TFE3* driven PEComas and tRCC, establishing a differential diagnosis may be important as well³, especially when they occur in the kidney, as the melanotic Xp11 tumors tend to be more aggressive with poorer prognosis, while the overall outcome of Xp11 translocation RCC is significantly better than that of the mesenchymal counterpart. We have now added **Supplementary Table S13**, which compares and contrasts tRCC and PEComas and modified the Discussion to reflect the continuum between the neoplasms **on page 25 (lines 585-590)**.

Reviewer #5 (Remarks to the Author):

References

- 1 Marletta, S. *et al.* SFPQ::TFE3-rearranged PEComa: Differences and analogies with renal cell carcinoma carrying the same translocation. *Pathol Res Pract* **270**, 155963 (2025). <https://doi.org/10.1016/j.prp.2025.155963>
- 2 Tretiakova, M. S. Chameleon TFE3-translocation RCC and How Gene Partners Can Change Morphology: Accurate Diagnosis Using Contemporary Modalities. *Adv Anat Pathol* **29**, 131-140 (2022). <https://doi.org/10.1097/pap.0000000000000332>
- 3 Wang, X. T. *et al.* SFPQ/PSF-TFE3 renal cell carcinoma: a clinicopathologic study emphasizing extended morphology and reviewing the differences between SFPQ-TFE3 RCC and the corresponding mesenchymal neoplasm despite an identical gene fusion. *Hum Pathol* **63**, 190-200 (2017). <https://doi.org/10.1016/j.humpath.2017.02.022>
- 4 Sanfilippo, R. *et al.* Role of Chemotherapy, VEGFR Inhibitors, and mTOR Inhibitors in Advanced Perivascular Epithelioid Cell Tumors (PEComas). *Clin Cancer Res* **25**, 5295-5300 (2019). <https://doi.org/10.1158/1078-0432.Ccr-19-0288>
- 5 Malinowska, I. *et al.* Perivascular epithelioid cell tumors (PEComas) harboring TFE3 gene rearrangements lack the TSC2 alterations characteristic of conventional PEComas: further evidence for a biological distinction. *The American journal of surgical pathology* **36**, 783-784 (2012). <https://doi.org/10.1097/PAS.0b013e31824a8a37>
- 6 Wagner, A. J. *et al.* Clinical activity of mTOR inhibition with sirolimus in malignant perivascular epithelioid cell tumors: targeting the pathogenic activation of mTORC1 in tumors. *J Clin Oncol* **28**, 835-840 (2010). <https://doi.org/10.1200/jco.2009.25.2981>
- 7 Di Malta, C. *et al.* Transcriptional activation of RagD GTPase controls mTORC1 and promotes cancer growth. *Science (New York, N.Y.)* **356**, 1188-1192 (2017). <https://doi.org/10.1126/science.aag2553>